EMBO
Molecular Medicine

# SK4 K[+] channels are therapeutic targets for the treatment of cardiac arrhythmias

Shiraz Haron-Khun[1,2,†], David Weisbrod[1,†], Hanna Bueno[1,†], Dor Yadin[2,†], Joachim Behar[3], Asher Peretz[1], Ofer Binah[4], Edith Hochhauser[5], Michael Eldar[2], Yael Yaniv[3], Michael Arad[2,*] (iD) & Bernard Attali[1,**] (iD)

## Abstract

Catecholaminergic polymorphic ventricular tachycardia (CPVT) is a stress-provoked ventricular arrhythmia, which also manifests sinoatrial node (SAN) dysfunction. We recently showed that SK4 calcium-activated potassium channels are important for automaticity of cardiomyocytes derived from human embryonic stem cells. Here SK4 channels were identified in human induced pluripotent stem cell-derived cardiomyocytes (hiPSC-CMs) from healthy and CPVT2 patients bearing a mutation in calsequestrin 2 (CASQ2-D307H) and in SAN cells from WT and CASQ2-D307H knock-in (KI) mice. TRAM-34, a selective blocker of SK4 channels, prominently reduced delayed afterdepolarizations and arrhythmic Ca[2+] transients observed following application of the β-adrenergic agonist isoproterenol in CPVT2-derived hiPSC-CMs and in SAN cells from KI mice. Strikingly, *in vivo* ECG recording showed that intraperitoneal injection of the SK4 channel blockers, TRAM-34 or clotrimazole, greatly reduced the arrhythmic features of CASQ2-D307H KI and CASQ2 knockout mice at rest and following exercise. This work demonstrates the critical role of SK4 Ca[2+]-activated K[+] channels in adult pacemaker function, making them promising therapeutic targets for the treatment of cardiac ventricular arrhythmias such as CPVT.

**Keywords** cardiac arrhythmia; catecholaminergic polymorphic ventricular tachycardia; pacemaker; potassium channel; SK4

**Subject Categories** Cardiovascular System; Genetics, Gene Therapy & Genetic Disease

## Introduction

Catecholaminergic polymorphic ventricular tachycardia (CPVT) is an inherited arrhythmogenic syndrome characterized by physical or emotional stress-induced polymorphic ventricular tachycardia in otherwise structurally normal hearts with a high fatal event rate in untreated patients (Priori *et al*, 2001; Hayashi *et al*, 2009; Priori & Chen, 2011; Abriel & Zaklyazminskaya, 2013). CPVT comprises heterogeneous genetic diseases, including mutations in ryanodine receptor type 2 (RyR2), calsequestrin 2 (CASQ2), triadin, or calmodulin (Leenhardt *et al*, 1995; Lahat *et al*, 2001; Priori *et al*, 2002; Chopra & Knollmann, 2011; Nof *et al*, 2011; Arad *et al*, 2012; Hwang *et al*, 2014). The RyR2 mutations (CPVT1) are "gain-of-function" mutations while CASQ2 mutants (CPVT2) are "loss-of-function" mutations, which both lead to diastolic Ca[2+] leakage from the sarcoplasmic reticulum (SR). This eventually produces local increases in cytosolic Ca[2+] that is extruded via the Na[+]–Ca[2+] exchanger NCX1 generating local depolarization with early or delayed afterdepolarizations (EADs or DADs) that trigger premature beats and fatal polymorphic ventricular tachycardia (Priori & Chen, 2011). Recent studies performed in human induced pluripotent stem cell-derived cardiomyocytes (hiPSC-CMs) from CPVT patients bearing mutations in either CASQ2 (D307H) or RyR2 (M4109R) showed that β-adrenergic stimulation caused marked elevation in diastolic Ca[2+], DADs, and oscillatory prepotentials (Itzhaki *et al*, 2012; Novak *et al*, 2012, 2015). Sinus bradycardia was also described in CPVT patients and in CPVT mouse models, suggesting that sinoatrial node (SAN) dysfunction may reflect another primary defect caused by CPVT mutations (Leenhardt *et al*, 1995; Postma *et al*, 2005; Katz *et al*, 2010; Neco *et al*, 2012; Faggioni *et al*, 2014; Glukhov *et al*, 2015). We identified SK4 calcium-activated potassium channels (K$_{Ca}$3.1) as being involved in the pacemaker activity of cardiomyocytes derived from human embryonic stem cells (hESC-CMs) (Weisbrod *et al*, 2013). Here we asked whether SK4

1   Department of Physiology and Pharmacology, The Sackler Faculty of Medicine, Tel Aviv University, Tel Aviv, Israel
2   Leviev Heart Center, Sheba Medical Center, Tel Hashomer, Tel Aviv, Israel
3   Laboratory of Bioenergetic and Bioelectric Systems, Biomedical Engineering Faculty, Technion—Israel Institute of Technology, Haifa, Israel
4   Department of Physiology, Ruth & Bruce Rappaport Faculty of Medicine, Technion—Israel Institute of Technology, Haifa, Israel
5   The Cardiac Research Laboratory of the Department of Cardiothoracic Surgery, Felsenstein Medical Research Center, Rabin Medical Center, Tel Aviv University, Petah Tikva, Israel
    *Corresponding author. Tel: +972 3 5304560; E-mail: michael.arad@sheba.health.gov.il
    **Corresponding author. Tel: +972 3 6405116; E-mail: battali@post.tau.ac.il
    †These authors contributed equally to this work

channels are expressed in SAN and play a role in CPVT. SK4 currents were found in hiPSC-CMs from healthy and CPVT2 (CASQ2-D307H) patients and in SAN cells from WT and CASQ2-D307H knock-in (KI) mice. TRAM-34, a selective blocker of SK4 channels, markedly reduced the occurrence of DADs and abnormal $Ca^{2+}$ transients detected following exposure to the β-adrenergic agonist isoproterenol in CPVT2-derived hiPSC-CMs and in SAN cells from CASQ2-D307H KI mice. Intraperitoneal injection (20 mg/kg) of SK4 channel blockers, TRAM-34 or clotrimazole, elicited bradycardia and noticeably reduced the ECG arrhythmic features recorded *in vivo* from CASQ2-D307H KI and CASQ2 knockout (KO) mice at rest and following treadmill exercise. The results suggest that SK4 channels play a critical role in normal and CPVT diseased pacemaker function. Importantly, our data indicate that SK4 channel blockers could open new horizons in the management of CPVT patients' rhythm disorders.

## Results

### SK4 channels are expressed in hiPSC-CMs and their blockade reduces arrhythmias recorded in hiPSC-CMs derived from CPVT2 (CASQ2-D307H) patients

Since pacemaker dysfunction was described in CPVT patients and CPVT mouse models (Leenhardt *et al*, 1995; Postma *et al*, 2005; Katz *et al*, 2010; Neco *et al*, 2012; Faggioni *et al*, 2014; Glukhov *et al*, 2015), we examined whether SK4 channels are expressed in SAN and play a role in CPVT. We used single spontaneously beating hiPSC-CMs (25-day-old EBs) derived from normal (healthy) and CPVT2 patients carrying the CASQ2 D307H mutation (Novak *et al*, 2012) and investigated their spontaneous firing and ionic currents. A voltage ramp was applied as previously (Weisbrod *et al*, 2013) and cells were held at −20 mV to substantially inactivate voltage-gated $Na^+$ and $Ca^{2+}$ currents (Fig 1A and B). In the absence of blockers (black traces), the voltage ramp revealed the presence of one and occasionally two inward humps peaking at about −40 mV and −5 mV and reflecting activation of residual T-type and L-type $Ca^{2+}$ currents, respectively. These inward humps vanished following exposure to 300 μM $CdCl_2$. Exposing cells to solution 1 (300 μM $CdCl_2$, 25 μM ZD7288, and 10 μM E-4031) suppressed the inward humps, shifted the reversal potential ($E_{rev}$) to the left, and markedly depressed inward and outward currents (orange trace). Addition of the selective SK4 channel blocker TRAM-34 (1 μM) to solution 1 decreased the ramp currents (green trace) (Fig 1A and B). Subtracting the ramp currents in solution 1 to those in solution 1 + TRAM-34 (1 μM) yielded the TRAM-34-sensitive current. Figure 1D shows the average traces of the TRAM-34-sensitive currents (using 1 μM TRAM-34) of normal and CPVT2-derived hiPSC-CMs, which mainly exhibited an outward component. Yet, small residual inward currents likely corresponding to cationic conductances were not fully blocked by solution 1 and therefore shifted the $E_{rev}$ to values more positive than those of $E_K$. TRAM-34-sensitive currents were never detected in zero internal free $Ca^{2+}$. Similar TRAM-34-sensitive current densities were found using either 1 or 5 μM TRAM-34 (Fig 1C). No significant differences were found in TRAM-34-sensitive current densities of normal and CPVT2 hiPSC-CMs (Fig 1C). For selectivity purposes, we examined whether TRAM-34 interfered

with major pacemaker currents in hESC-CMs. We found that 5 μM TRAM-34 did not alter T-type and L-type $Ca^{2+}$ currents measured by the two inward humps (zero free $Ca^{2+}$ in pipette solution; Appendix Fig S1A). While 25 μM ZD7288 blocked $I_f$ at all voltages (~70% inhibition at −100 mV), 5 μM TRAM-34 did not affect the $I_f$ current at any voltage. The NCX blocker KB-R7943 (3 μM) potently inhibited the NCX current, but 5 μM TRAM-34 was ineffective (Appendix Fig S1B and C). SK4 channel expression was confirmed at the protein level, where an SK4 immunoreactive band of about 50 kDa was identified in Western blots from beating cluster lysates of both normal and CPVT2 hiPSC-CMs (Fig 1E).

Exposure of normal hiPSC-CMs to 100 nM isoproterenol significantly increased the firing rate and the slope of diastolic depolarization (DD). Adding 1 μM TRAM-34 to the isoproterenol solution significantly depolarized the maximal diastolic potential (MDP) and decreased the firing rate and the DD slope, which eventually culminated by a suppression of the pacing (Fig 2A and B). Similar experiments were performed on CASQ2 D307H hiPSC-CMs. Isoproterenol did not significantly increase the beating rate on CPVT2 hiPSC-CMs, but instead, it triggered DADs (Fig 2C, arrows). Strikingly, adding 1 μM TRAM-34 to the isoproterenol solution drastically reduced the number of DADs and led to subsequent and reversible cessation of the spontaneous activity (Fig 2C and D).

### SK4 channels are expressed in SAN cells and their inhibition lessens the arrhythmic phenotype of SAN cells from CASQ2-D307H KI mice

Individual SAN cells were isolated from WT and CASQ2-D307H homozygous KI mice (Song *et al*, 2007; Katz *et al*, 2010) and recorded as described above, except that cells were held at −40 mV to improve their stability. In the absence of blockers (black traces), the voltage ramp revealed the presence of one inward hump peaking at about −40 mV and reflecting activation of T-type $Ca^{2+}$ currents with minor contribution of L-type $Ca^{2+}$ currents (Fig 3A and B). Upon exposure of cells to solution 1 (orange traces), the inward hump and substantial ramp currents disappeared. Addition of 1 μM TRAM-34 to solution 1 (green traces) decreased the outward ramp currents. Like for hiPSC-CMs, while the average traces of the TRAM-34-sensitive currents (using 1 μM TRAM-34) of WT and CASQ2-D307H SAN cells exhibited a prominent outward component, small residual inward currents that were not completely blocked by solution 1 shifted the $E_{rev}$ to values more positive than those of $E_K$ (Fig 3D). Similar TRAM-34-sensitive current densities were found using either 1 or 5 μM TRAM-34 (Fig 3C). Comparable densities of TRAM-34-sensitive currents were isolated in SAN cells from WT and CASQ2-D307H KI mice (Fig 3C). Confirming the expression of SK4 channels and CASQ2 in adult mouse heart of WT and CASQ2-D307H KI mice, Western blots of lysates from SAN, right and left atrial appendages, and right and left ventricles showed specific immunoreactive bands corresponding to SK4 channel and to CASQ2 protein (Fig 3E). Quantitative analysis of the blots showed no significant differences in the heart tissues between the WT and CASQ2-D307H KI mice (Fig 3F).

Next, we recorded the spontaneous activity of isolated SAN cells. Exposure of WT SAN cells to 2 μM clotrimazole, another SK4 channel blocker, significantly decreased the firing rate and the DD slope (Appendix Fig S2B, violet trace). These effects were reversible during washout (Appendix Fig S2B, blue trace). Similarly, 2 μM

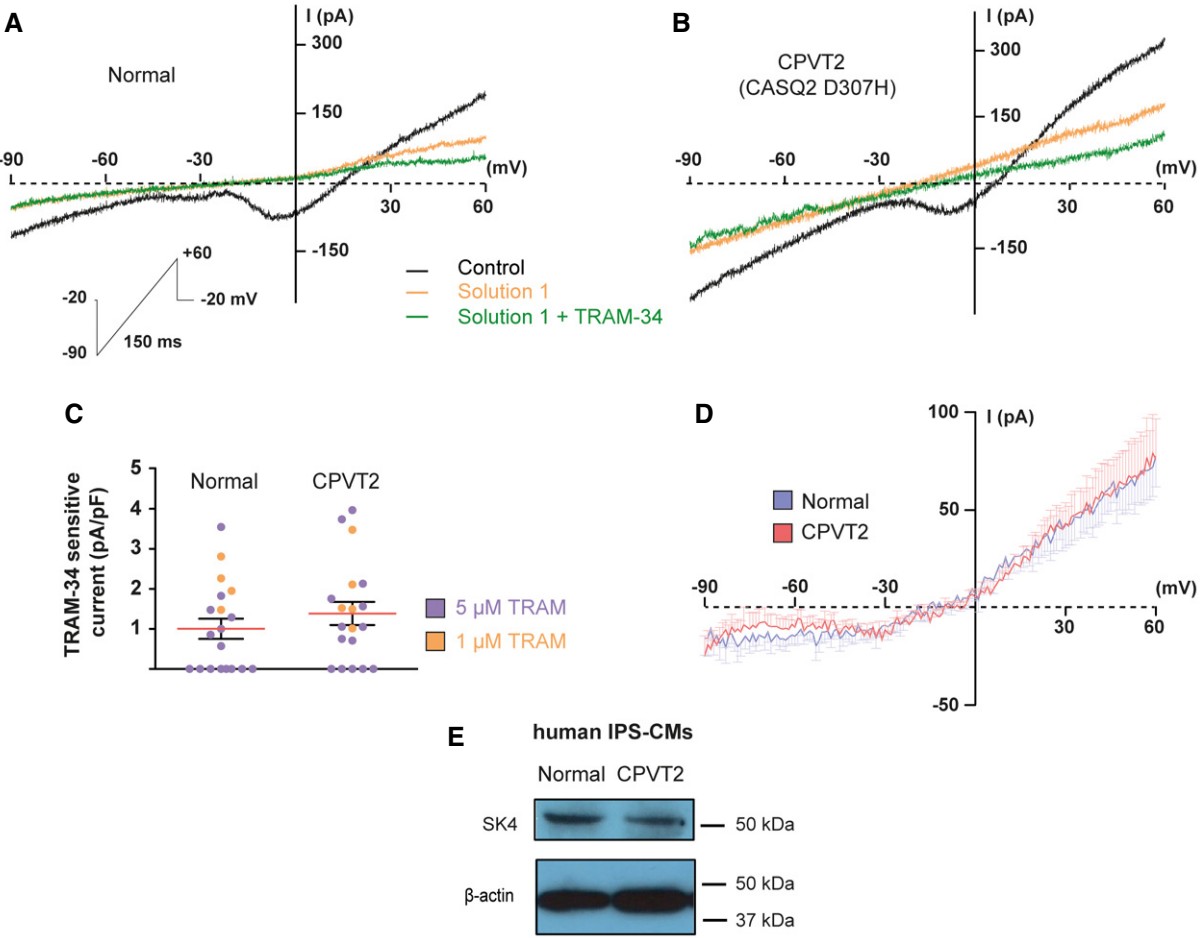

**Figure 1.  SK4 channels are expressed in hiPSC-CMs derived from a healthy normal individual and a CPVT2 (CASQ2-D307) patient.**

A   Representative traces of an hiPSC-CM derived from a healthy normal individual following a voltage ramp under the indicated conditions. Solution 1 included 300 μM CdCl₂, 25 μM ZD7288, and 10 μM E-4031.

B   Representative traces of an hiPSC-CM derived from a CPVT2 (CASQ2 D307H) patient.

C   Scatter plot of the TRAM-sensitive current densities measured at +60 mV with 1 or 5 μM TRAM-34. Current densities were $1.00 \pm 0.25$ pA/pF in normal ($n = 19$) and $1.39 \pm 0.29$ pA/pF in CPVT2 ($n = 18$). Not statistically different (two-tailed unpaired $t$-test).

D   Average traces of the TRAM-34-sensitive currents using 1 μM TRAM-34 of normal ($n = 4$) and CPVT2-derived hiPSC-CMs ($n = 5$). For clarity, the SEM bars are shown for every mV.

E   Representative Western blots of beating EB lysates from a normal individual and a CPVT2 (CASQ2 D307H) patient showing immunoreactive SK4 protein (≈50 KDa).

TRAM-34 decreased the spontaneous beating rate and depolarized the MDP before cessation of the pacing (Appendix Fig S2C and D). Isoproterenol (50 nM) significantly increased the pacing of SAN cells from WT mice with an increased DD slope (Fig 4A and B). Adding 2 μM TRAM-34 to isoproterenol depolarized the MDP, markedly reduced the DD slope, decreased the beating rate, and eventually stopped the pacing activity in three out of seven cells. In SAN cells from CASQ2-D307H KI mice, addition of 50 nM isoproterenol initially produced a positive chronotropic effect. However, after 1–2 min isoproterenol led to DADs (Fig 4C, arrows). Remarkably, when TRAM-34 was added to the isoproterenol solution, the occurrence of DADs was drastically reduced (Fig 4C and D).

To investigate the spontaneous calcium transients of the SAN, we exposed to Fluo-4 AM intact SAN tissue preparations dissected *ex vivo* from WT and CASQ2-D307H KI mice as previously described (Torrente *et al*, 2015). In SAN from WT mice, the rate of calcium

transients was significantly increased in the presence of 100 nM isoproterenol and the additional exposure of 2 μM TRAM-34 did not alter the pattern of the Ca²⁺ waves (Fig 5A). Consistent with previous studies in different CPVT1 and CPVT2 mouse models and hiPSC-CMs (Itzhaki *et al*, 2012; Neco *et al*, 2012; Novak *et al*, 2012, 2015; Glukhov *et al*, 2015; Torrente *et al*, 2015), exposing SANs from CASQ2-D307H KI mice to 100 nM isoproterenol produced various Ca²⁺ transient abnormalities, which we classified according to their degree of severity (Fig 5B and C). In Fig 5C are shown local Ca²⁺ release (upper left), double-humped transients (upper right), large-stored released Ca²⁺ waves (lower left), and calcium alternans (lower right). Strikingly, adding 2 μM TRAM-34 normalized the shapes of isoproterenol-induced aberrant calcium waves in SAN from CASQ2-D307H KI mice (Fig 5B). For instance, TRAM-34 brought back to zero the number of SANs displaying double-humped transients or large-stored released Ca²⁺ waves (Fig 5D).

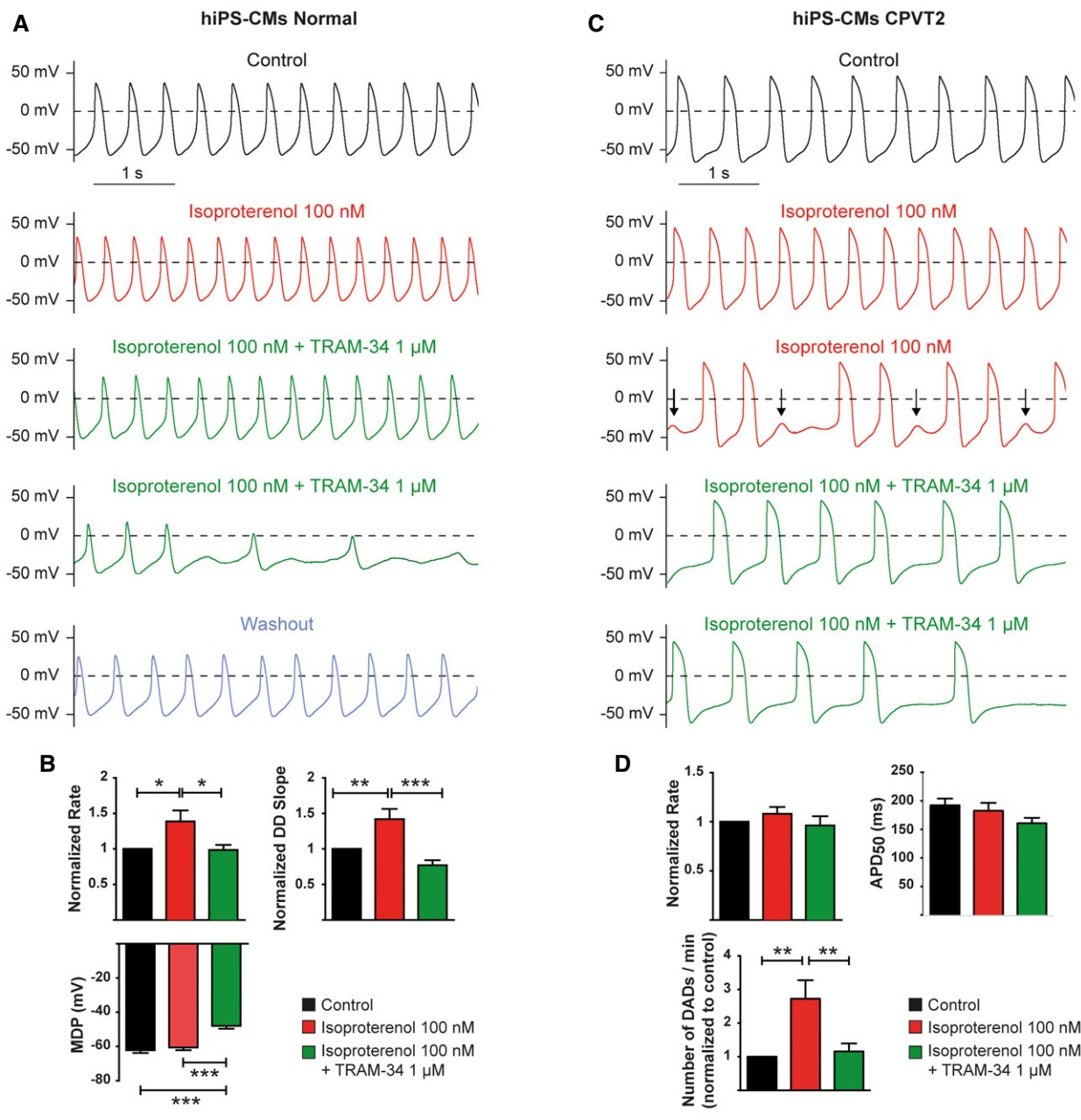

**Figure 2. Blockade of SK4 channels by 1 μM TRAM-34 reduces arrhythmias recorded in hiPSC-CMs derived from a CPVT2 (CASQ2-D307H) patient.**

A   Representative traces of spontaneous APs recorded in a hiPSC-CM derived from a normal individual under the indicated conditions.

B   Histograms of statistical data of the beating rate, DD slope, and MDP of hiPSC-CMs from a normal individual. One-way ANOVA followed by Tukey's multiple comparison test. For rate (normalized to Control), *$P < 0.05$, $n = 24$; for DD slope (normalized to Control), **$P < 0.01$ and ***$P < 0.0001$, $n = 24$; for MDP, ***$P < 0.0001$, $n = 24$. Bars and error bars are mean ± SEM.

C   Representative traces of spontaneous APs recorded in a hiPSC-CM derived from a CPVT2 patient under the indicated conditions.

D   Histograms of statistical data of the beating rate, APD$_{50}$, and DADs of hiPSC-CMs from a CPVT2 patient. One-way ANOVA followed by Tukey's multiple comparison test. For the rate, $P = $ ns, $n = 21$; for the APD$_{50}$, $P = $ ns, $n = 19$; for the DADs, **$P < 0.01$, $n = 19$. Bars and error bars are mean ± SEM.

## Blockade of SK4 channels improves *in vivo* the ECG arrhythmic features of CASQ2-D307H KI and CASQ2 KO mice

A heart telemetry device was implanted in WT, CASQ2-D307H KI, and CASQ2 KO mice for continuous ECG recording at rest and during treadmill exercise. For each session, continuous ECG recording was performed with the same animals receiving first intraperitoneal (IP) injection of vehicle (peanut oil) and then the SK4 channel blocker. TRAM-34 (20 mg/kg, IP) significantly decreased the resting heart rate of WT mice by 16 ± 3% as measured by the PP interval (Fig 6A and B). Interestingly, a significant prolongation of 20% in the PR interval was also seen on the

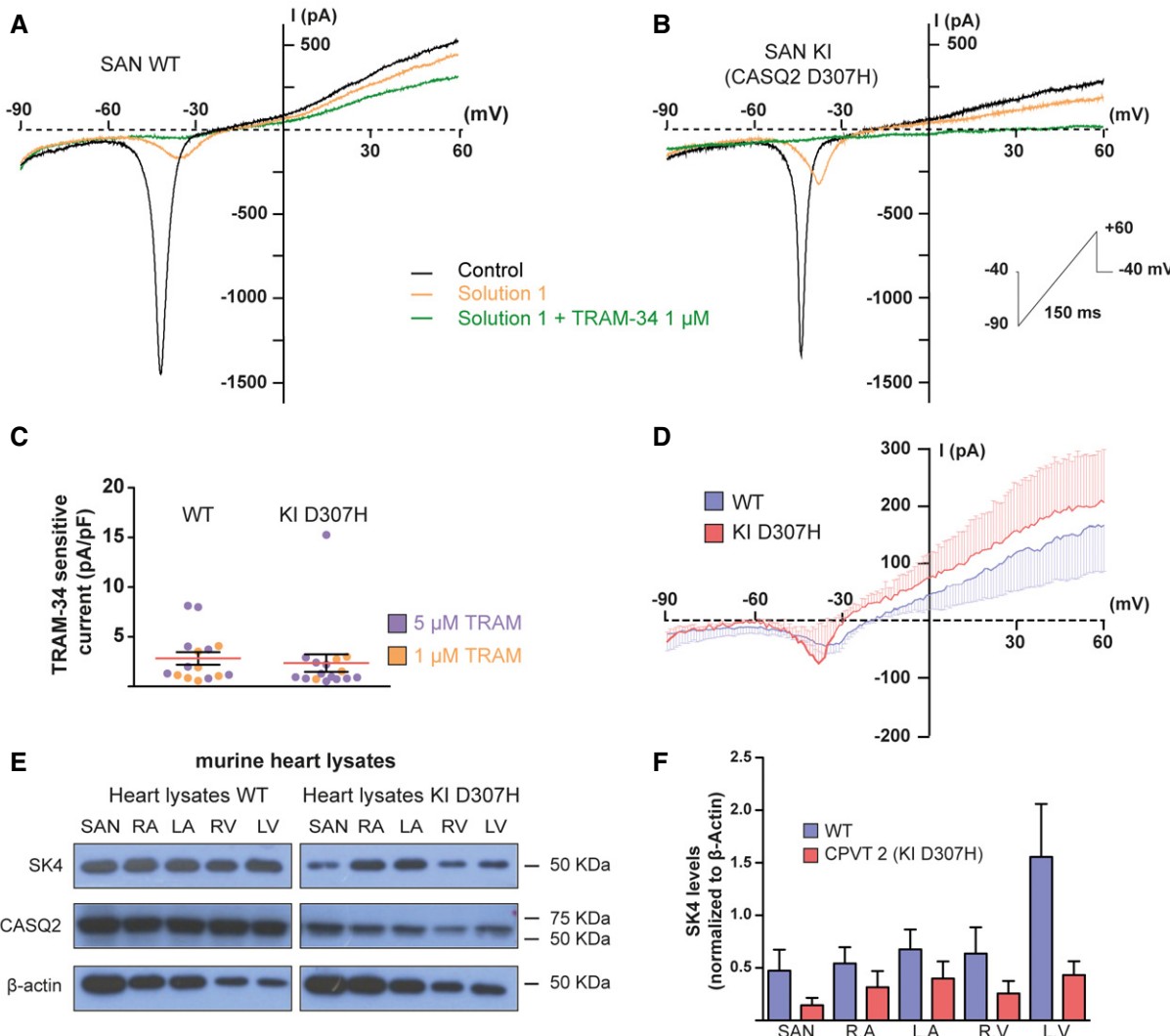

**Figure 3.  SK4 channels are expressed in SAN cells from WT and CASQ2-D307H KI mice.**

A    Representative traces of a SAN cell from WT mice following a voltage ramp under the indicated conditions.
B    Representative traces of a SAN cell from CASQ2-D307H KI mice.
C    Scatter plot of the TRAM-sensitive current densities measured at +60 mV with 1 or 5 μM TRAM-34. Current densities were 2.82 ± 0.63 pA/pF in WT (*n* = 15) and 2.36 ± 0.89 pA/pF in CASQ2-D307H KI mice (*n* = 16). Not statistically different (two-tailed unpaired *t*-test).
D    Average traces of the TRAM-34-sensitive currents using 1 μM TRAM-34 of WT (*n* = 7) and CASQ2-D307H KI mice (*n* = 4). For clarity, the SEM bars are shown for every mV.
E    Representative Western blots of heart lysates from WT and CASQ2-D307H KI mice showing the immunoreactive bands of SK4, CASQ2, and β-actin proteins in SAN, right and left atrial appendages, and right and left ventricles.
F    Quantification of the SK4 channel immunoreactive protein (normalized to β-actin) in different heart regions (*n* = 3). Not statistically different (two-tailed unpaired *t*-test). Error bars are SEM.

Source data are available online for this figure.

ECG traces of WT mice (Fig 6A and B). TRAM-34 produced similar bradycardic effects and PR interval prolongation during treadmill exercise of WT mice (Fig 7A and B). Confirming the importance of SK4 channels in the pacemaker function of adult WT mice, another SK4 channel blocker clotrimazole (20 mg/kg, IP) significantly reduced the resting heart rate by 16 ± 6% (Appendix Fig S3A and B) and prolonged by 27% the PR interval. A similar trend was noticeable during treadmill exercise (Appendix Fig S4A and B).

CASQ2-D307H KI and CASQ2 KO mice displayed lower basal heart rates compared to WT mice but also irregular sinus rhythm and ventricular premature complexes (VPCs) as shown on the ECG traces (Fig 6C–F). Frequently, these VPCs produced a desynchronization of the PQRS complexes, accompanied by variable P–Q intervals (Appendix Fig S3C; upper row, see arrows). Sometimes, the VPCs were so severe that the P waves were absent because there were absorbed into the premature QRS complexes (Fig 7E; upper row). TRAM-34 injection (20 mg/kg, IP) to these mice produced like

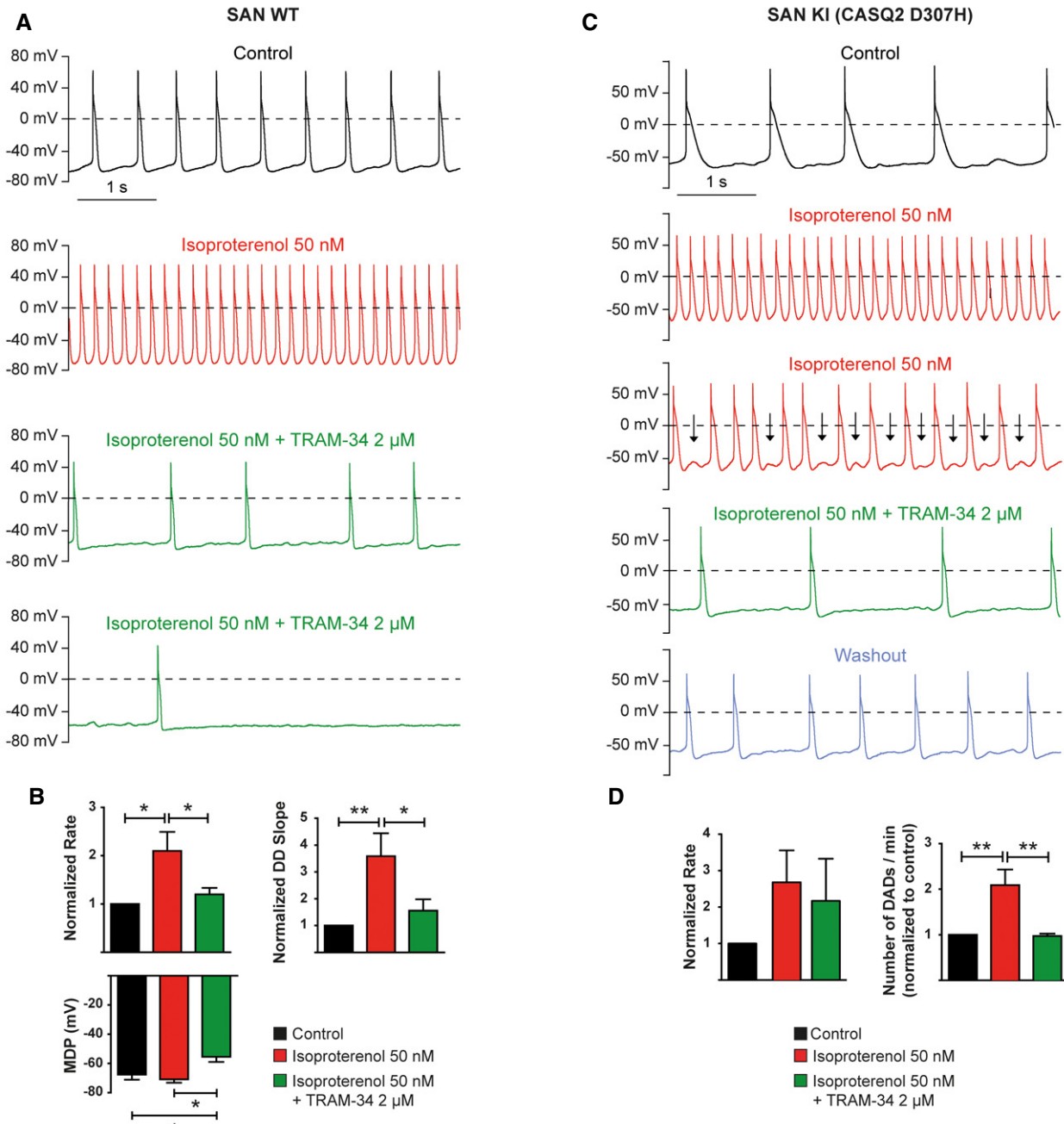

**Figure 4. Inhibition of SK4 channels by 2 μM TRAM-34 lessens the arrhythmic phenotype of SAN cells from CASQ2-D307H KI mice.**

A   Representative traces of spontaneous APs recorded in a SAN cell from WT mice under the indicated conditions.
B   Histograms of statistical data of the beating rate, DD slope, and MDP of SAN cells from WT mice. One-way ANOVA followed by Tukey's multiple comparison test. For rate, *P < 0.05, n = 6; for DD slope, *P < 0.05, **P < 0.01, n = 7; for MDP, *P < 0.05, n = 5. Bars and error bards are mean ± SEM.
C   Representative traces of spontaneous APs recorded in a SAN cell from CASQ2-D307H KI mice under the indicated conditions.
D   Histograms of statistical data of the beating rate, and MDP of SAN cells from CASQ2-D307H KI mice. One-way ANOVA followed by Tukey's multiple comparison test. For the rate, P is not significant, n = 5; for DADs, **P < 0.01, n = 5. Bars and error bards are mean ± SEM.

in WT animals significant bradycardic effects and PR prolongation (Fig 6D and F). Remarkably, TRAM-34 injection improved the ECG arrhythmic features observed under resting conditions and totally suppressed them in nine out of 12 KI mice. During treadmill exercise, the ECG cardiac abnormalities were aggravated with

"non-sustained" and even "sustained" ventricular tachycardia (Fig 7C and E). Under these conditions, TRAM-34 injection decreased the prevalence and severity of arrhythmias (Table 1). Notably, TRAM-34 was able to restore the P waves that disappeared because of the VPC-induced desynchronization of the PQRS complexes in

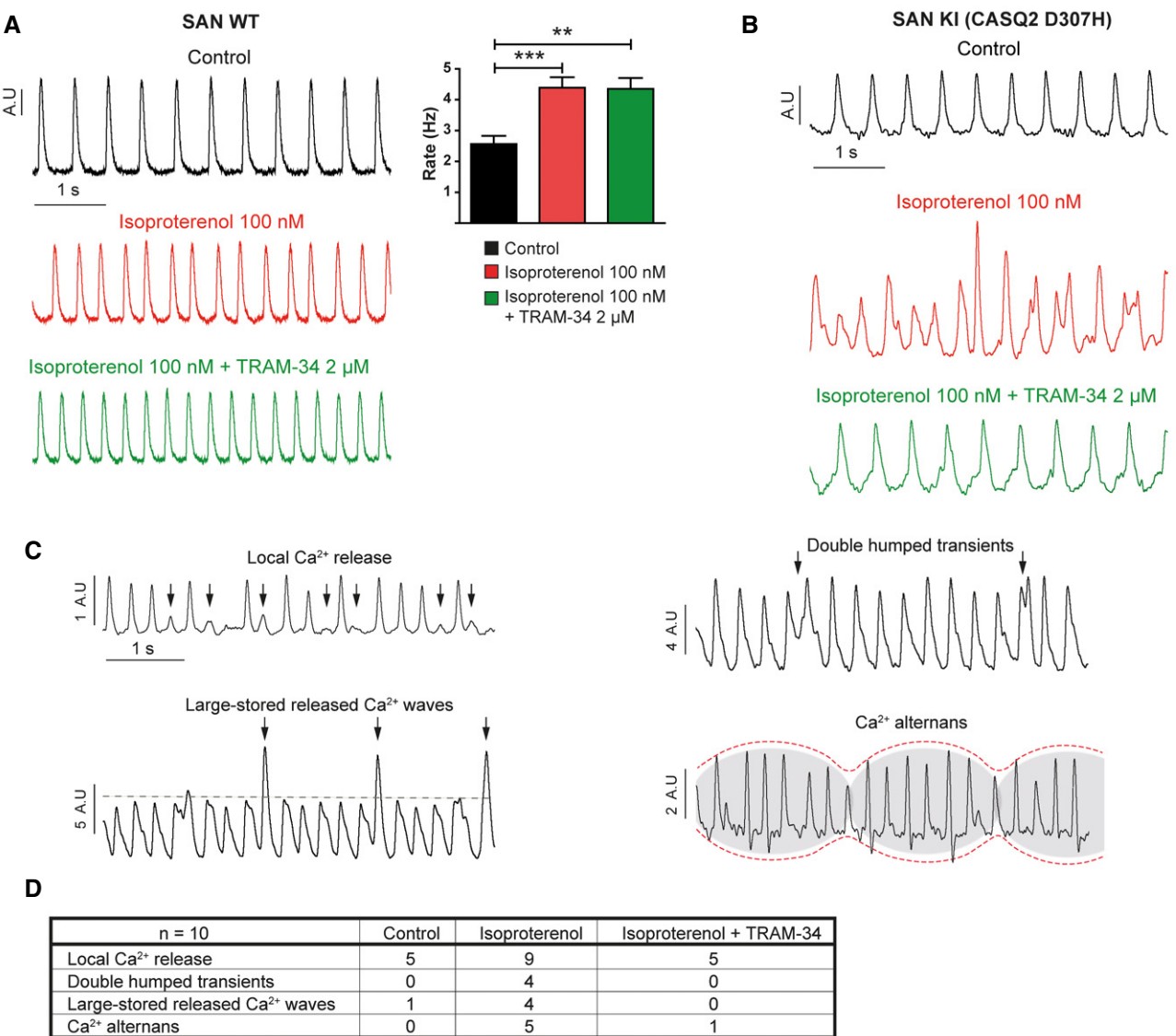

**Figure 5. Isoproterenol leads to abnormal SAN calcium transients, which are improved with TRAM-34.**

A   Left, representative traces of spontaneous calcium transients recorded *ex vivo* in intact SAN tissue preparations from WT mice under the indicated conditions. Right: data summary of calcium transient rate (one-way ANOVA: **$P < 0.01$, ***$P < 0.001$, $n = 12$). Bars and error bars are mean $\pm$ SEM.

B   Representative traces of spontaneous calcium transients recorded from intact SAN of CASQ2 D307H KI mice.

C   Representative traces of different types of calcium transient abnormalities recorded in intact SAN from CASQ2 D307H KI mice, termed as "local Ca²⁺ release", "double-humped transients", "large-stored released Ca²⁺ waves", and "calcium alternans".

D   Data summary of the arrhythmic calcium transients in SAN from CASQ2 D307H KI under the indicated conditions.

CASQ2 KO animals (Fig 7E). During treadmill exercise, TRAM-34 also produced significant sinus bradycardia and PR interval prolongation in KI and KO mice (Fig 7C–F). Clotrimazole (20 mg/kg, IP) elicited similar effects to those observed with TRAM-34. Under basal conditions (Appendix Fig S3) and during treadmill exercise (Appendix Fig S4), bradycardia and PR prolongation were noticed in CASQ2-D307H KI and CASQ2 KO mice following clotrimazole injection (Appendix Figs S3 and S4). Importantly, clotrimazole improved the ECG arrhythmic features observed at rest and following treadmill exercise and even succeeded to convert them to normal sinus rhythm in three out of five KI mice and four out of six KO mice at rest (Appendix Figs S3 and S4, and Table 1). Clotrimazole was capable of resynchronizing the disorganized PQRS complexes in CASQ2-D307H KI mice (Appendix Fig S3C; see arrows).

## Mathematical modeling of SK4 channels in mouse SAN

To further explore the mechanistic insight into how SK4 currents ($I_{SK4}$) contribute to SAN pacemaker activity, the impact of $I_{SK4}$ in SAN firing rate was examined using mathematical modeling, where $I_{SK4}$ was added to the mouse model implemented by Kharche *et al* (2011). Appendix Fig S5 shows the model predictions with and

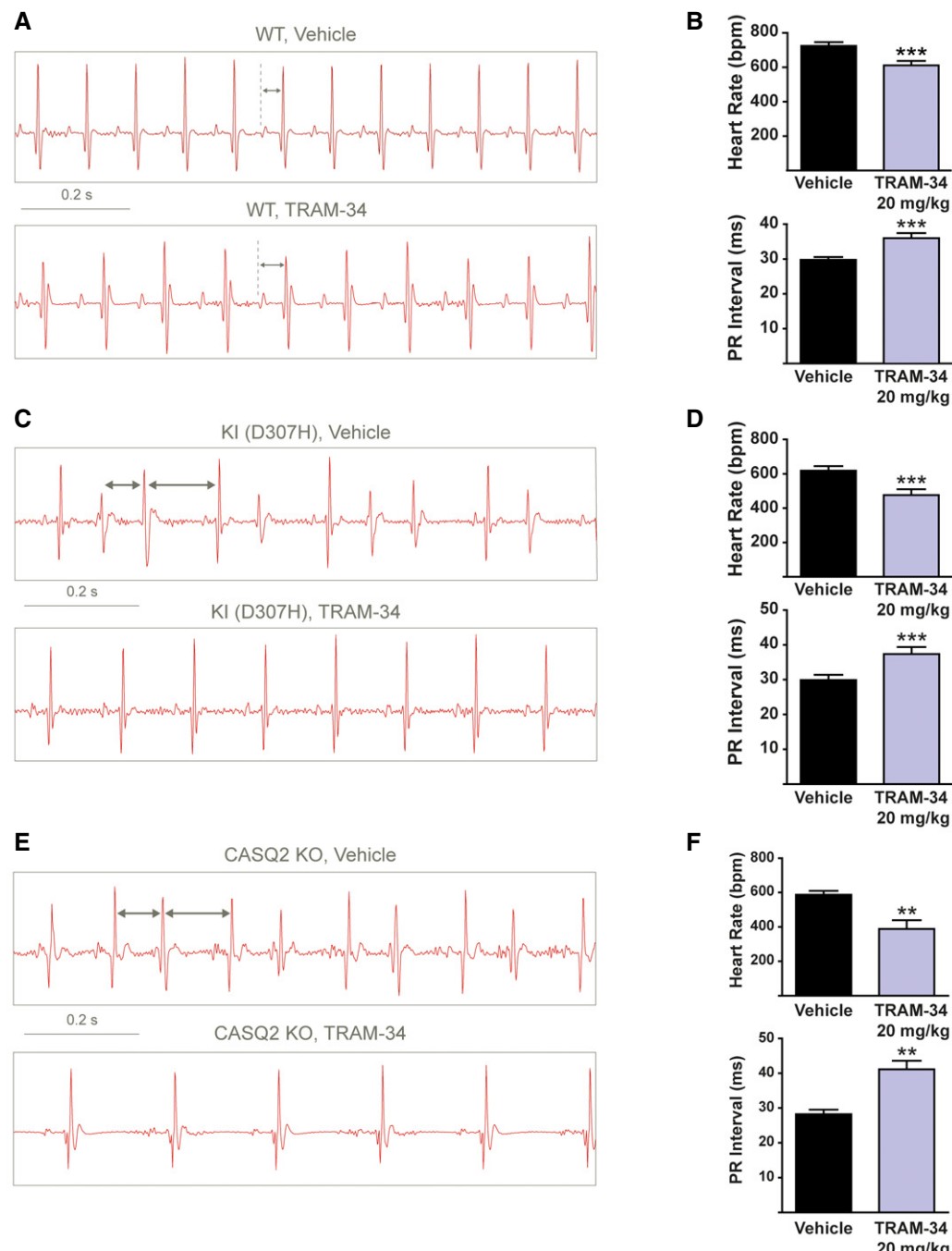

**Figure 6. Blockade of SK4 channels by TRAM-34 improves the ECG arrhythmic features of CASQ2-D307H KI and CASQ2 KO mice under rest conditions.**

A    Representative ECG recording following IP injection of vehicle (upper) and 20 mg/kg TRAM-34 (lower) in WT mice at rest. Sequential vehicle and TRAM-34 injections were performed on the same animal.

B    Data summary of heart rate (paired t-test; ***P = 0.0003, n = 10) and PR interval (paired t-test; ***P = 0.0004, n = 10) in WT mice at rest. Error bars: ± SEM.

C    Representative ECG recording following IP injection of vehicle (upper) and 20 mg/kg TRAM-34 (lower) in CASQ2-D307H KI mice at rest.

D    Data summary of heart rate (paired t-test; ***P < 0.0001, n = 12) and PR interval (paired t-test; ***P < 0.0001, n = 12) in CASQ2-D307H KI mice at rest. Error bars: ± SEM.

E    Representative ECG recording following IP injection of vehicle (upper) and 20 mg/kg TRAM-34 (lower) in CASQ2 KO mice at rest.

F    Data summary of heart rate (paired t-test; **P = 0.004, n = 7 mice) and PR interval (paired t-test; **P = 0.004, n = 7) in CASQ2 KO mice at rest. Error bars: ± SEM.

without the contribution of $I_{SK4}$. From the $Ca^{2+}$-dependent sensitivity curve of SK4 channel activation measured by Logsdon et al (1997), we constrained the model with a Hill slope of $n_x = 2.7$ and a

$Ca^{2+}$ dissociation constant of $k_x = 0.27$ μM. Assuming activation and deactivation time constants to $\tau_a = 5$ ms and $\tau_d = 50$ ms, respectively, as referred for all SK channels (Berkefeld et al, 2010),

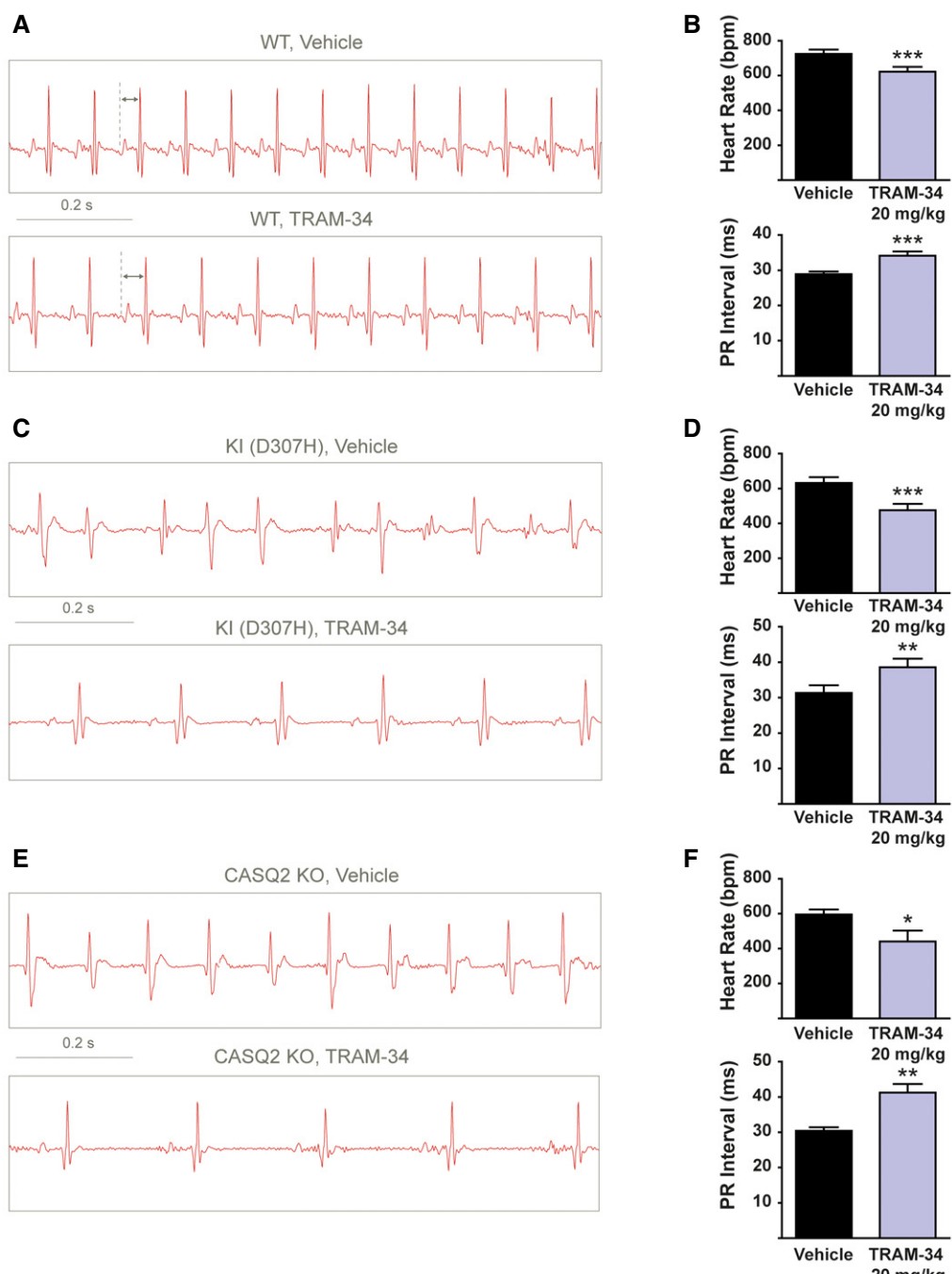

**Figure 7. Blockade of SK4 channels by TRAM-34 improves the ECG arrhythmic features of CASQ2-D307H KI and CASQ2 KO mice during treadmill exercise.**

A   Representative ECG recording following intraperitoneal injection of vehicle (upper) and 20 mg/kg TRAM-34 (lower) in WT mice during treadmill exercise.

B   Data summary of heart rate (paired *t*-test; ***$P = 0.001$, $n = 10$) and PR interval (paired *t*-test; ***$P = 0.0005$, $n = 10$) in WT mice during exercise. Error bars: $\pm$ SEM.

C   Representative ECG recording following IP injection of vehicle (upper) and 20 mg/kg TRAM-34 (lower) in CASQ2-D307H KI mice during treadmill exercise.

D   Data summary of heart rate (paired *t*-test; ***$P = 0.0004$, $n = 11$) and PR interval (paired *t*-test; **$P = 0.0099$, $n = 11$) in CASQ2-D307H KI mice during exercise. Error bars: $\pm$ SEM.

E   Representative ECG recording following IP injection of vehicle (upper) and 20 mg/kg TRAM-34 (lower) in CASQ2 KO mice during treadmill exercise.

F   Data summary of heart rate (paired *t*-test; *$P = 0.0165$, $n = 7$) and PR interval (paired *t*-test; **$P = 0.0042$, $n = 7$) in CASQ2 KO mice during exercise. Error bars: $\pm$ SEM.

addition of $I_{SK4}$ resulted in a slower AP upstroke in late DD (leading to a decrease in the firing rate) and in a faster AP repolarization (leading to an increase in the firing rate). Because of this time delay in channel activation and deactivation processes, the net effect of adding $I_{SK4}$ to the model resulted in an increase in the firing rate (Appendix Fig S5A and B). $I_{SK4}$ is still active even after calcium

**Table 1.  Improvement of arrhythmogenic features at rest or during exercise in CPVT2 CASQ2-D307H KI and CASQ2 KO mice after IP injection of the SK4 blockers TRAM-34 or clotrimazole.**

| Number of mice (n) | 5 Vehicle | 5 Clotrimazole 20 mg/kg | 12 Vehicle | 12 TRAM-34 20 mg/kg |
|---|---|---|---|---|
| **KI at rest** | | | | |
| Normal | 0 | 3 | 3 | 9 |
| VPC | 4 | 2 | 6 | 3 |
| NSVT | 1 | 0 | 3 | 0 |
| SVT | 0 | 0 | 0 | 0 |
| **KI during exercise** | | | | |
| Normal | 0 | 0 | 1 | 4 |
| VPC | 1 | 2 | 2 | 4 |
| NSVT | 4 | 2 | 9 | 4 |
| SVT | 0 | 1 | 0 | 0 |

| Number of mice (n) | 6 Vehicle | 6 Clotrimazole 20 mg/kg | 6 Vehicle | 6 TRAM-34 20 mg/kg |
|---|---|---|---|---|
| **KO at rest** | | | | |
| Normal | 0 | 4 | 1 | 6 |
| VPC | 4 | 2 | 5 | 0 |
| NSVT | 2 | 0 | 0 | 0 |
| SVT | 0 | 0 | 0 | 0 |
| **KO during exercise** | | | | |
| Normal | 0 | 0 | 0 | 3 |
| VPC | 0 | 2 | 0 | 0 |
| NSVT | 5 | 3 | 3 | 3 |
| SVT | 1 | 1 | 3 | 0 |

VPC, ventricular premature complexes; NSVT, non-sustained ventricular tachycardia; SVT, sustained ventricular tachycardia. The types of arrhythmic features were classified following their severity: sinusal rhythm (normal), ventricular premature contractions (VPC), NSVT, and SVT.

concentration in the membrane subspace $[Ca^{2+}]_{sub}$ returned to its basal value (Appendix Fig S5C and D). This is mainly due to the time constant of the deactivation process. By keeping the values of $n_x = 2.7$ and $k_x = 0.27$ μM, but removing from the equation the activation and deactivation time constants, the effect of adding $I_{SK4}$ was opposed to the experimental findings as the firing rate decreased (Appendix Fig S5E and F). Because no time delay occurs in the activation and deactivation processes, the contribution of $I_{SK4}$ during late DD outweighs its effect during late repolarization. This leads $I_{SK4}$ to vanish very quickly after the peak of the AP and to follow the $[Ca^{2+}]_{sub}$ trajectory (Appendix Fig S5G and H).

## Discussion

This study demonstrates the pivotal role of SK4 $Ca^{2+}$-activated $K^+$ channels in adult pacemaker function, making them promising

therapeutic targets for the treatment of cardiac ventricular arrhythmias such as CPVT. Until recently, the presence and function of SK4 channels in the heart were overlooked. We originally identified SK4 channels in human embryonic stem cell-derived cardiomyocytes and showed that they play a crucial role in human embryonic cardiac automaticity (Weisbrod *et al*, 2013). Other laboratories found that SK4 channels are critical for cardiac pacemaker fate determination in embryonic stem cells and induced pluripotent stem cells of mice (Kleger *et al*, 2010; Kleger & Liebau, 2011; Liebau *et al*, 2011) and humans (Muller *et al*, 2012). Treatment with an SK4 channel opener was found to differentiate mouse embryonic stem cells into cardiomyocytes with a strong enrichment of pacemaker-like cells. This differentiation was accompanied by induction of SAN-specific genes and by a loss of the ventricular-specific gene program (Kleger *et al*, 2010).

Since CPVT patients exhibit pacemaker dysfunction and CPVT mouse models display defects in SAN automaticity (Leenhardt *et al*, 1995; Postma *et al*, 2005; Katz *et al*, 2010; Neco *et al*, 2012; Faggioni *et al*, 2014; Glukhov *et al*, 2015), we explored whether SK4 channels are expressed in adult SAN and play a role in CPVT. In the present work, we provide the first evidence that SK4 channels are not only expressed in spontaneously beating hESC-CMs and hiPSC-CMs but in SAN cells too. Inhibition of SK4 $K^+$ currents by TRAM-34 reduced the intrinsic SAN firing rate. Our data reveal that in SAN cells, SK4 channels are novel regulators of mouse SAN automaticity. Cardiac automaticity is achieved by the integration of voltage-gated currents (membrane clock) with rhythmic $Ca^{2+}$ release from internal $Ca^{2+}$ stores ($Ca^{2+}$ clock) (Seyama, 1976; Brown, 1982; Shibasaki, 1987; Sanguinetti & Jurkiewicz, 1990; Hagiwara *et al*, 1992; DiFrancesco, 1993, 2010; Guo *et al*, 1997; Huser *et al*, 2000; Vinogradova *et al*, 2002; Mangoni & Nargeot, 2008; Lakatta & DiFrancesco, 2009; Lakatta *et al*, 2010). SAN pacemaker activity is due to the ability to generate DD, where a cohort of inward currents slowly depolarize the membrane potential until reaching the threshold of a next action potential (AP) mainly triggered by opening of voltage-gated $Ca^{2+}$ channels. These include funny currents ($I_f$), T-type $Ca^{2+}$ currents, and the $Na^+/Ca^{2+}$ exchanger NCX1 that is activated in its forward mode by cyclical SR $Ca^{2+}$ release via RyR2 (Huser *et al*, 2000; Vinogradova *et al*, 2002). Outward $K^+$ currents can affect very differently murine SAN excitability. While $I_{KR}$, SK2, and $I_{to}$ repolarize the AP, $I_{KACh}$ (GIRK4) can act during DD to dampen SAN firing rate (Xu *et al*, 2003; Mangoni & Nargeot, 2008; Li *et al*, 2009; Mahida, 2014). Our data clearly indicate that SK4 channels do not significantly alter AP duration but affect the MDP and the DD slope (Figs 2 and 4, and Appendix Fig S2). In all SK channels, activation results from $Ca^{2+}$ binding to calmodulin followed by conformational changes that open the pore. The time constant ($\tau_a = 5$ ms) of this activation process was shown to be strongly dependent on intracellular $Ca^{2+}$ (Berkefeld *et al*, 2010). SK channel deactivation, initiated by dissociation of $Ca^{2+}$, is independent of intracellular $Ca^{2+}$ and occurs on a much slower time-scale ($\tau_d = 15–60$ ms). SK channels can remain active for more than 100 ms after $[Ca^{2+}]_i$ has returned to resting levels (Berkefeld *et al*, 2010). To fit our numerical modeling with the experimental data, we needed to constrain the activation and deactivation time constants to $\tau_a = 5$ ms and $\tau_d = 50$ ms, respectively. Precisely because of the slow deactivation time process, the net effect of adding $I_{SK4}$ to the mouse SAN model developed by Kharche *et al*

(2011) resulted in an increase in the pacemaker rate. Under these conditions, the model showed that $I_{SK4}$ is still active even after calcium concentration in the subspace $[Ca^{2+}]_{sub}$ returned to its basal value (Appendix Fig S5). The model prediction clearly indicated that removing from the equation the activation and deactivation time constants yields results that do not match to the experimental data. Because of this slow channel deactivation, we suggest that SK4 channel contribution becomes significant only at the late repolarization, thereby contributing to the MDP hyperpolarization, which facilitates activation of $I_f$ and recovery from inactivation of voltage-gated $Ca^{2+}$ channels. Thus, the net effect of SK4 channel activation will be an increase in the firing rate. SK4 channels may act in SAN like $BK_{Ca}$ channels in hippocampal neurons, where their activation counterintuitively increases excitability, while their inhibition reduces firing (Gu *et al*, 2007). We predict that activation of SK4 channels will increase the SAN pacing rate and their blockade will reduce it. Our *in vitro* and *in vivo* data obtained with the SK4 channel blockers, TRAM-34 and clotrimazole, on the pacing rate of isolated SAN cells and on ECG parameters of WT mice are in excellent agreement with this assumption. Both blockers produced significant bradycardic effects during rest and following treadmill exercise. An indirect impact of TRAM-34 or clotrimazole on autonomic input to SA and AV nodes *in vivo* can be excluded because both blockers exert similar effects on isolated SAN cells. In line with these data, RA-2, a structurally different molecule from TRAM-34 and clotrimazole, with a mixed blocker activity toward SK4 and SK2 channels, induced bradycardia in mice, an effect abolished in SK4 knockout mice (Olivan-Viguera *et al*, 2015).

The prolongation of the PR interval is usually related to either AV node or the His–Purkinje system and suggests that SK4 channels are expressed in the conduction system. However, the PR interval represents a composite of several components. A prolonged PR interval can also reflect delayed interatrial conduction times. Prolonged PR interval was often considered detrimental to diastolic filling because it leads to a decrease in diastolic filling time. However, a prolonged PR interval could be also beneficial, because it may allow for complete atrial emptying during the atrial systole. Interestingly, previous transcriptional analysis showed a ninefold upregulation of SK4 in the developing conduction system compared to SK1–3 (Horsthuis *et al*, 2009).

Reflecting functional redundancy among SAN ionic conductances, it is interesting to notice that additional $Ca^{2+}$-activated $K^+$ channels have been characterized in the murine cardiac pacemaker. Blockade of SK2 channels prolonged the AP duration in atrioventricular nodal cells and knockout of SK2 channels in mice resulted in bradycardia and prolongation of the PR interval (Zhang *et al*, 2008). Conversely, overexpression of SK2 channels decreased AP duration, increased spontaneous firing rate of atrioventricular nodal cells, and reduced PR and RR intervals in ECG (Zhang *et al*, 2008). More recently, $Ca^{2+}$- and voltage-activated BK $K^+$ channels were also identified in murine SAN cells (Lai *et al*, 2014). Genetic ablation or pharmacological inhibition of BK channels were associated with reduced heart rate in ECG and slowed SAN cells pacing without alteration of AP duration (Lai *et al*, 2014). This apparent redundancy of $Ca^{2+}$-activated $K^+$ currents indicates that they share similar properties such as bradycardia upon channel blockade (SK2, SK4, and BK), but they also exhibit subtle differences notably regarding their impact on AP duration (e.g., SK2 versus SK4).

Probably, thanks to the multiplicity and intrinsic redundancy of ion channels in the cardiac pacemaker, our work showed that inhibition of SK4 $K^+$ channels rescues *in vitro* the cardiac arrhythmias exhibited by hiPSC-CMs derived from CPVT2 patients carrying the CASQ2 D307H mutation and by SAN cells isolated from CASQ2-D307H KI mice. Hence, TRAM-34 markedly reduced the occurrence of DADs and abnormal $Ca^{2+}$ transients detected following exposure to the β-adrenergic agonist isoproterenol. Notably, SK4 channel blockers could protect *in vivo* the animals from deleterious ventricular arrhythmic features revealed by ECG in CASQ2-D307H KI and CASQ2 KO mice at rest and after treadmill exercise. TRAM-34 and clotrimazole were able to restore the P waves that disappeared following the VPC-induced desynchronization of the PQRS complexes. VPCs, non-sustained ventricular tachycardia (NSVT), and sustained ventricular tachycardia were significantly reduced following a single IP injection (20 mg/kg) of clotrimazole or TRAM-34. The SK4 channel blockers protected the CASQ2-D307H KI and CASQ2 KO mice from harmful polymorphic ventricular tachycardia without being pro-arrhythmic by themselves, since neither sinus arrest nor second-order AV block was recorded in the animals, including WT mice. Despite the blockade of SK4 channels, the functional redundancy of $Ca^{2+}$-activated $K^+$ channels likely preserves the delicate balance of inward and outward currents necessary for normal pacemaking. Along the same line, recent studies showed that cardiac SAN arrhythmias induced by silencing either HCN4 ($I_f$ current) or Cav1.3 (L-type $Ca^{2+}$ currents) could be rescued by genetic deletion or pharmacological inhibition of GIRK4 channels ($I_{KACh}$ currents) (Lai *et al*, 2014; Mesirca *et al*, 2014, 2016). Thanks to their bradycardic effect and slowed AV conduction, but also to their impact on the MDP, SK4 channel blockers could be beneficial for preventing ventricular tachycardia by prolonging the refractory period.

Therapies for CPVT are phenotype driven and include exercise prohibition and β1-adrenergic blockade. The response to β-blockers is incomplete and often declines with time because of an escape phenomenon (Priori *et al*, 2002; Hayashi *et al*, 2009). The options in unresponsive patients include additional drugs, primarily flecainide, or implanting a defibrillator (ICD) and sympathetic denervation (Van der Werf *et al*, 2012). Although very effective in mice, $Ca^{2+}$ channel blockers have a limited benefit in humans, even when combined with β-blockers. Our data indicate that SK4 $K^+$ channels are novel, promising therapeutic targets for the treatment of cardiac ventricular arrhythmias. Importantly, pharmacological inhibitors of SK4 channels already exist and are developed for therapy of sickle-cell anemia, asthma, autoimmune encephalomyelitis, immunosuppression, and ischemic stroke (Wulff & Kohler, 2013). This work suggests that the therapeutic indication of SK4 channel blockers could be extended to ventricular tachyarrhythmias in CPVT and possibly in other arrhythmic pathologies of different etiologies such as the long QT syndrome.

# Materials and Methods

### Animals

*SvEv* mice (3–6 months old) homozygous for the CASQ2 D307H mutation [CASQ2 D307H knock-in (KI)] or for the off-frame exon 9

deletion [CASQ2$\Delta$/$\Delta$ knockout (KO)] and matched wild-type (WT) mice were used in this study (Song et al, 2007). Mice were maintained and bred in a pathogen-free facility on regular rodent chow with free access to water and 12-h light and dark cycles. The procedures followed for experimentation and maintenance of the animals were approved by the Animal Research Ethics Committee of Tel Aviv University (M-14-063) in accordance with Israeli law and in accordance with the Guide for the Care and Use of Laboratory Animals (1996, National Academy of Sciences, Washington, DC, USA).

### Human induced pluripotent stem cell culture and cardiac differentiation

Human induced pluripotent stem cells (hiPSC) derived from normal healthy individuals and from patients bearing the CASQ2 D307H mutation (CPVT2) were grown on mitomycin C-inactivated mouse embryonic fibroblasts (MEF), in order to maintain them in an undifferentiated state (Novak et al, 2012, 2015). The cells were maintained pluripotent in a culture medium containing 80% DMEM F-12 (Biological Industries), 20% Knockout SR (Invitrogen), 2 mM L-glutamine, 0.1 mM β-mercaptoethanol (Gibco), and 1% NEA (Gibco), supplemented with 4 ng/ml bFGF (Invitrogen). The medium was replaced daily until the colonies were ready to passage (every 4–5 days). For EB induction (d0), hiPSC colonies were removed from their MEF feeder by collagenase IV treatment and collected. After centrifugation, the cells were re-suspended in EB medium containing 80% DMEM (Gibco), 20% FBS (Biological Industries), 1% NEA, and 1 mM L-glutamine and plated on 58-mm Petri dishes. After 7 days of culture in suspension, EBs were plated on 0.1% gelatin-coated plates and checked daily until a spontaneous beating activity was visible. Because CASQ2 is lately expressed in hiPSC-CMs, 25-day-old EBs were used (Novak et al, 2012, 2015). The beating clusters were mechanically dissected from EBs, following a three-step dissociation protocol (Novak et al, 2012; Weisbrod et al, 2013). The hiPSC-CMs were isolated and plated on Matrigel-coated glass coverslips (13 mm diameter) in 24-well plates. The coverslips were then incubated at 37°C, and a recovery period of 2 days was given before any electrophysiological experiment was performed.

### Mouse SAN dissection and cell dissociation

WT and CASQ2 D307H KI mice were anesthetized with isofluorane and sacrificed by cervical dislocation. The heart was rapidly removed and transferred into Tyrode solution containing heparin. After the atria were pined and the superior and inferior vena cava localized, the ventricles were removed. The SAN was anatomically identified between the superior and inferior vena cava, the crista terminalis, and the interatrial septum. The area was cleaned, cut into small strips, and washed into a low-calcium solution containing (in mM) 140 NaCl, 5.4 KCl, 0.5 MgCl$_2$, 1.2 KH$_2$PO$_4$, 5 HEPES-NaOH, 50 taurine, 5.5 glucose (pH 6.9). The osmolarity was adjusted if needed to 315 mOsm. The same solution supplemented with 1 mg/ml albumin, 200 μM CaCl$_2$, collagenase type I (Worthington) or liberase TH (Roche), protease (Sigma), and elastase (Sigma) was used for enzymatic digestion as described (Mesirca et al, 2014). In this step, the tissue was gently re-suspended with a polished Pasteur pipette in this solution for 9–13 min at 37°C. SAN samples were then washed three times in a modified "Kraftbrühe" solution

containing (in mM) 70 glutamic acid, 80 KOH, 20 KCl, 10 γ-hydroxy-butyric acid sodium salt, 10 KH$_2$PO$_4$, 10 HEPES-KOH, 10 taurine, 1 mg/ml albumin, 0.1 EGTA-KOH (pH 7.2). The same solution was used to re-suspend the single cells with a pipette by vigorous up and down shaking, between 3 and 5 min at 37°C. Cells were then gradually exposed to increasing concentrations of calcium, following a "Ca$^{2+}$ readaptation" protocol (Mesirca et al, 2014). Experiments were performed the same day at 33°C.

### Drugs

Isoproterenol, clotrimazole, and E-4031 were purchased from Sigma, while ZD-7288 and TRAM-34 were from Tocris. For in vivo telemetric recordings, Tram-34 was solubilized into peanut oil, while clotrimazole was prepared in peanut oil supplemented with 1% ethanol.

### Electrophysiology

In all experiments, the coverslips were perfused at 33°C with an external solution containing (in mM) 140 NaCl, 4 KCl, 11 glucose, 1.2 MgCl$_2$, 1.8 CaCl$_2$, 5.5 HEPES titrated to pH 7.4 with NaOH and adjusted at 320 mOsm with sucrose. Whole-cell patch-clamp recordings were performed with an Axopatch 700B amplifier (Molecular Devices) and pCLAMP 10.5 software (Molecular Devices). Signals were digitized at 5 kHz and filtered at 2 kHz. Microelectrodes with resistances of 4–7 MΩ were pulled from borosilicate glass capillaries (Harvard Apparatus) and filled with an intracellular solution containing (in mM) 130 KCl, 5 MgATP, 5 EGTA, 10 HEPES titrated to pH 7.3 with KOH and adjusted at 290 mOsm with sucrose. Unless otherwise stated, internal free calcium concentrations were 100 and 1 μM for current-clamp and voltage-clamp experiments, respectively, and were titrated with EGTA and CaCl$_2$ using the MaxChelator software (www.stanford.edu/~cpatton/maxc.html). The spontaneous automaticity of isolated ∼ SAN cells was recorded under perforated-patch conditions by adding 30 μM β-escin (Mesirca et al, 2014) to the intracellular solution containing (in mM) 130 KCl, 10 NaCl, 10 HEPES, 0.2 EGTA-KOH, 2 MgATP, 6.6 phosphocreatine, 0.05 cAMP, and 1 μM free Ca$^{2+}$ (pH 7.2). To record SK4 K$^+$ current, a voltage ramp protocol was applied. SAN and hiPSC-CMs were held at −40 and −20 mV, respectively, to substantially inactivate voltage-gated Na$^+$ and Ca$^{2+}$ currents. Cells were stepped from −90 mV to +60 mV for 150 ms. Then, a cocktail (solution 1) containing (in mM) 0.3 cadmium, 0.025 ZD-7288, and 0.01 E-4031 was applied extracellularly to inhibit residual L-type and T-type voltage-gated Ca$^{2+}$ currents, I$_{f,}$and the IKr currents, respectively. Subsequently, TRAM-34 (5 μM) was added to solution 1 to inhibit SK4 K$^+$ currents, which were defined as TRAM-34-sensitive currents. For voltage-clamp recording of SAN cells, the intracellular solution was the same to that described above for recording spontaneous automaticity.

### Calcium transient measurements

SAN tissue preparations were dissected ex vivo from WT and CASQ2-D307H KI mice as previously described (Torrente et al, 2015). The dissected whole SAN tissue was pinned on a handmade chamber and was incubated in a Tyrode solution containing 10 μM

Fluo-4 AM (Thermo Fisher Scientific) and pluronic acid for 1 h at 37°C in the dark. The SAN tissue was washed in Tyrode at 37°C in the dark for 10 min before experiments. Fluorescence of calcium transients was recorded using a photomultiplier (PTi D-104) at 35°C, and the analog signals were digitized using Digidata 1440 (Molecular Devices) and analyzed with pCLAMP 10.5 software.

## Western blotting

Mouse atrial and ventricular tissues were cut into small pieces (left and right atrial appendages, left and right ventricles, sinoatrial node) or beating clusters from normal and CASQ2-D307H hiPSC-CMs were re-suspended in ice-cold lysis buffer [50 mM Tris–HCl pH 7.5, 100 mM NaCl, 1% Nonidet P-40, 0.1% SDS, supplemented with protease cocktail inhibitor (Sigma-Aldrich) and 1 mM phenyl-methylsulfonyl fluoride (Sigma-Aldrich)], incubated on ice for 45 min, shaken by vortex every 2–3 min, and centrifuged for 15 min at 4°C at 16,000 g. Equal amounts of proteins (30 μg) of the resulting lysate supernatant were mixed with Laemmli sample buffer and fractionated by 10% SDS–PAGE. The resolved proteins were electro-blotted onto a nitrocellulose membrane. The membrane was incubated with the primary antibodies followed by horseradish peroxidase-conjugated secondary anti-IgG antibodies (1:10,000). The primary antibodies were diluted into 5% skim milk-TBST (Tris-buffered saline, 0.1% Tween-20). The mouse anti-SK4/KCa3.1 (SAB1409264 Sigma 1:1,000) was used for rodent lysates and the rabbit anti-SK4/KCa3.1 (AV35098 Sigma 1:2,500) was used for human hiPSC-CMs lysates. Both SK4 antibodies were incubated overnight at 4°C. The rabbit anti-Casq2 (18422-1-AP Proteintech, 1:2,500) and the mouse monoclonal anti β-actin (MP Biomedical clone C4 691001 1:10,000) were incubated 1 h at room temperature. Signals were developed using SuperSignal West Pico Chemiluminescent Substrate (Thermo Scientific).

## *In vivo* telemetric recordings

Telemetric ambulatory long-term ECG recordings, analogous to Holter monitoring in humans, were obtained with implantable transmitters. The investigator was blinded for the mice genotypes. WT, CASQ2-D307H KI, and CASQ2 KO *SvEv* mice were anesthetized with ketamine (75–90 mg/kg) and xylazine (5–8 mg/kg) intraperitoneally (IP) (Kepro, Holland), and a midline incision was made along the spine. An implantable 3.5 g wireless radiofrequency transmitter (DSI MM USA, device weight 3.8 g) was aseptically inserted into a subcutaneous tissue pocket in the back as previously described (Katz *et al*, 2010; Kurtzwald-Josefson *et al*, 2014). Animals were allowed to recover after surgery 5–6 days before any experiments. Baseline electrocardiograms (ECG) were obtained 15 min after IP injection of the appropriate vehicle (peanut oil or peanut oil supplemented with ethanol 1%). For pharmacological experiments, the same mouse was used a few hours after baseline ECG recordings (vehicle injection) and for subsequent ECG recordings upon IP injection of 20 mg/kg clotrimazole or TRAM-34. Telemetered ECG tracings were obtained in conscious mice at rest for one minute and during peak exercise (i.e., the first minute of recovery). In the treadmill exercise, mice were forced to exercise on a rodent treadmill, gradually increasing the speed up to a maximum of 15 m/min. Ventricular tachycardia (VT) was defined as four or more consecutive

### The paper explained

#### Problem

Catecholaminergic polymorphic ventricular tachycardia is an inherited arrhythmogenic syndrome, which is characterized by physical or emotional stress-induced ventricular tachycardia in otherwise structurally normal hearts with a high fatal event rate in untreated patients. CPVT is one of the most malignant cardiac channelopathies, which also manifests sinoatrial node dysfunction. We originally identified SK4 Ca$^{2+}$-activated potassium channels in human embryonic stem cell-derived cardiomyocytes and showed that they play a crucial role in human embryonic cardiac automaticity. Because CPVT patients exhibit pacemaker dysfunction and CPVT mouse models display defects in sinoatrial node automaticity, we explored whether SK4 potassium channels are expressed in sinoatrial node and play a role in CPVT.

#### Results

As experimental model, we used pacemaker cells derived from human induced pluripotent stem cells (hiPSC-CMs) of healthy and CPVT2 patients bearing a mutation in calsequestrin 2 (CASQ2-D307H). We also used adult sinoatrial node cells from WT and CASQ2-D307H knock-in (KI) mice. TRAM-34, a selective blocker of SK4 potassium channels, prominently reduced the DADs and the arrhythmic Ca$^{2+}$ transients observed following application of the β-adrenergic agonist isoproterenol in CPVT2-derived hiPSC-CMs and in sinoatrial node cells from KI mice. Strikingly, *in vivo* ECG recording showed that intraperitoneal injection (20 mg/kg) of the SK4 channel blockers, TRAM-34 or clotrimazole, greatly reduced the ventricular arrhythmic features of CASQ2-D307H KI and CASQ2 knockout mice at rest and following exercise. TRAM-34 and clotrimazole were able to restore the P waves that disappeared following the ventricular premature complex-induced desynchronization of the PQRS complexes.

#### Impact

This study demonstrates for the first time the pivotal role of SK4 potassium channels in adult pacemaker function. Therapies for CPVT include exercise prohibition and β1-adrenergic blockade. However, the response to β1-blockers is incomplete and often declines with time because of an escape phenomenon. This work suggests that SK4 channel blockers could be of therapeutic help for ventricular tachyarrhythmias in CPVT and possibly for other ventricular arrhythmias of different etiologies such as the long QT syndrome.

ventricular beats. If this phenotype was consecutively observed for more than 15 s, it was defined as "sustained" ventricular tachycardia (SVT). Shorter VTs were characterized as "non-sustained" (NSVT). All other ventricular arrhythmias, such as premature beats, ventricular bigeminy, couplets, and triplets, were all defined as ventricular premature contractions (VPCs) (Katz *et al*, 2010).

## Data analysis

Rate, AP duration at 50% of repolarization (APD$_{50}$), delayed afterdepolarizations (DADs), current densities, and calcium transients were analyzed with the Clampfit program (pClamp 10.5; Molecular Devices). Leak subtraction was performed offline using the Clampfit software. Sinus rhythm, PR interval, and ECG arrhythmic features were analyzed with the LabChart 8 Reader (ADInstruments). Data were analyzed with Excel (Microsoft) and Prism 5.0 (GraphPad Software) and are expressed as mean ± SEM. Statistical analysis was performed using the two-tailed paired Student's *t*-test and the linear

regression for correlation or by one-way ANOVA followed by Tukey's multiple comparison test. *P*-values of < 0.05 were assumed significant.

### Mathematical model

All details about the mathematical model are described in the Appendix.

**Expanded View** for this article is available online.

### Acknowledgements

This work was supported by a grant from the Israel Science Foundation (763/10) to MA, (ISF 292/13) to OB and (ISF 1215/13 and 2092/14) and the Fields Fund for Cardiovascular Research to BA. BA holds the Andy Libach Professorial Chair in clinical pharmacology and toxicology. We thank Prof. Dario DiFrancesco, Dr. Andrea Barbuti and Manuel Paina (The Pacelab, Milano), and Dr. Pietro Mesirca (IGF-CNRS, Montpellier) for their kind training and advices in isolating the mouse sinoatrial node preparation. We are grateful to Profs. Jonathan G. Seidman and Christine E. Seidman for initially providing us with the *SvEv* mice.

### Author contributions

SHK, DW, HB, and AP performed the electrophysiology and biochemical experiments in hiPSC-CMs and SAN cells and analyzed the data. DY and EH performed the *in vivo* heart telemetric experiments and analyzed the data. JB and YY performed the numerical modeling. OB, ME, MA, and BA designed the work and analyzed data. DW, YY, MA, and BA wrote the manuscript.

### Conflict of interest

The authors declare that they have no conflict of interest.

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
