## [Review Process File · EMBO Molecular Medicine]

Manuscript EMM-2016-06937

SK4 K⁺ channels are therapeutic targets for the treatment of cardiac arrhythmias

Shiraz Haron-Khun, David Weisbrod, Hanna Bueno, Dor Yadin, Joachim Behar, Asher Peretz, Ofer Binah, Edith Hochhauser, Michael Eldar, Yael Yaniv, Michael Arad, Bernard Attali

Corresponding author: Bernard Attali, Tel Aviv University & Michael Arad, Sheba Medical Center

Review timeline:

Submission date:	08 August 2016
Editorial Decision:	09 September 2016
Revision received:	10 January 2017
Editorial Decision:	20 January 2017
Revision received:	23 January 2017
Accepted:	25 January 2017

Editor: Roberto Buccione

Transaction Report:

1st Editorial Decision

09 September 2016

Thank you for the submission of your manuscript to EMBO Molecular Medicine. We are sorry that it has taken longer than usual to get back to you on your manuscript. We have now heard back from the three reviewers who were asked to evaluate your manuscript.

As you will see, although #2 is largely positive, in aggregate a number of concerns are raised that require your action. I will not go into detail, as their comments are quite clear.

Reviewer 1 mentions the need for further mechanistic insight into how SK4 channel block rescues the electrical properties of cardiac cells, to strengthen the findings and increase their impact. S/he also points to a number of deficiencies in data processing and presentation, which are also shared in part by reviewers 2 and 3. I wish to add that during our reviewer cross-commenting exercise, #2 agreed that the concerns raised by #1 and 3 required appropriate action.

We agree on all points and, as mentioned by reviewer 1, since you have the data available, these should be incorporated into the current manuscript, at least to the extent indicated by the reviewer.

In conclusion, while publication of the paper cannot be considered at this stage, we would be pleased to consider a revised submission, with the understanding that the Reviewers' concerns must be addressed in full including with additional experimental data where appropriate and that acceptance of the manuscript will entail a second round of review.

Please note that it is EMBO Molecular Medicine policy to allow a single round of revision only and that, therefore, acceptance or rejection of the manuscript will depend on the completeness of your responses included in the next, final version of the manuscript.

As you know, EMBO Molecular Medicine has a "scooping protection" policy, whereby similar findings that are published by others during review or revision are not a criterion for rejection. However, I do ask you to get in touch with us after three months if you have not completed your revision, to update us on the status. Please also contact us as soon as possible if similar work is published elsewhere.

Please note that EMBO Molecular Medicine now requires a complete author checklist (<http://embomolmed.embopress.org/authorguide#editorial3>) to be submitted with all revised manuscripts. Provision of the author checklist is mandatory at revision stage; The checklist is designed to enhance and standardize reporting of key information in research papers and to support reanalysis and repetition of experiments by the community. The list covers key information for figure panels and captions and focuses on statistics, the reporting of reagents, animal models and human subject-derived data, as well as guidance to optimise data accessibility. The Author checklist will be published alongside the paper, in case of acceptance, within the transparent review process file.

Finally, we now mandate that all corresponding authors list an ORCID digital identifier. You may do so through our web platform upon submission and the procedure takes <90 seconds to complete. We also encourage co-authors to supply an ORCID identifier, which will be linked to their name for unambiguous name identification.

I look forward to seeing a revised form of your manuscript as soon as possible.

***** Reviewer's comments *****

Referee #1 (Remarks):

The study by Haron-Khun et al. describes SK4 block as a novel putative therapeutic strategy to treat Catecholaminergic Polymorphic Ventricular Tachycardia (CPVT). Therefore the authors studied iPSCs and Knock-In mouse models of CPVT and particularly focused on the role of SK4 sino-atrial node (SAN) cells and the effects of SK blockers in mouse ECG recordings. Concerning the mechanism of action for the rescue of the CPVT phenotype by SK4 channel block the authors state: Because of this slow channel deactivation, we suggest that SK4 channel contribution becomes significant only at the late repolarization, thereby contributing to the MDP hyperpolarization, which facilitates activation of I_f and recovery from inactivation of voltage gated Ca^{2+} channels. Thus, the net effect of SK4 channel activation will be an increase in the firing rate (manuscript in preparation). As this not provided data describes the molecular mechanism of action and the principle how SK4 channel block alters/rescues the electrical properties of cardiac cells, some of the data must be moved into the current EMBO Mol Med manuscript to undermine the counter intuitive idea that sino-atrial SK4 expression is in fact increasing excitability.

Additional major points

Fig. 1. The TRAM-sensitive current was present in 7 out of 15 (control) and 9 out of 13 cells (CPVT). As this data describes the SK4 current, the analyses is not done careful enough. The average TRAM-sensitive current density (Fig. 1c) should include all cells and not only the cells in which the experiments apparently worked and the authors isolated a relative large TRAM-sensitive current. In addition, the authors should illustrate the TRAM-sensitive current. Here the average of the TRAM-sensitive current would make sense to be illustrated in Fig. 1a and Fig. 1b. This additional data will show the rectification properties and reversal potential of the putative SK4 current.

Why is the control data not illustrated for the DD slope in Fig. 1e, while it is always provided (for Rate, APD50, Number of DADs..)?

Fig. 1e: The authors state "Adding 5 μ M TRAM-34 depolarized the maximal diastolic potential (MDP)". As this is the major mechanism of action, as argued in the Discussion section (see above), the authors should provide the data and statistics in Fig. 1e.

Fig. 2a-c. There are problems with the analyses of data display of the TRAM-sensitive current in SAN cells. The representative traces in Fig. 1a suggest a huge TRAM-sensitive current in comparison to the recordings of the SAN KI (Fig. 1b). This cannot be representative as the analyses in Fig. 1c suggest that there is no difference in current densities. As for Fig. 1, the average TRAM-sensitive currents should be displayed. Moreover no information is provided whether also these cells, as for the iPSCs, did not always have a TRAM-sensitive current. As stated above the statistics should be done using all cells and not only the "responders" to get an impression of the SK4 current amplitude in this cardiac tissue.

Also Fig. 2e lacks the analyses of the maximal diastolic potential.

The authors should consider in the Discussion/Results section that a PQ interval prolongation could be also caused by atrial effects and does not strictly indicate a role of SK4 in the conduction system

Referee #2 (Comments on Novelty/Model System):

It is a very nice paper, which has strong translational potential given that CPVT-related arrhythmias are currently mostly treated by conventional treatments such as beta-blocker or ICD implantation. I am sure that a clinical study is underway to clarify whether SK4 treatment is a viable novel therapeutic concept. Given however the widespread expression in the heart, SK4 blockade may also be detrimental in certain conditions. Nonetheless an important paper of the pioneering lab that implicated SK4 channel into pacemaking.

Referee #2 (Remarks):

This is a very nice paper implicating SK4 channels as therapeutic target in CPVT-related arrhythmias. The authors of the manuscript should be congratulated for their very nice paper. Nonetheless I have a few remarks that should be addressed.

1. In the Western blot shown in Fig. 2d it appears that the expression levels of SK4 are affected by the CPVT phenotype and are reduced in the SAN and left and right ventricles. Is this impression correct and have you quantified your Western blots? If that is indeed the case, it would indicate that the CPVT disease process may have impact on SK4 channel expression.
2. The slowing of the PR interval after TRAM-34 blockade suggests that not only is there expression in the SAN but also in the AV node. In the discussion you propose that expression is present throughout the cardiac conduction tissue. This could be demonstrated by performing immunohistochemical stainings provided that your antibody not only works in Western blots but also on tissue sections. Based on the Western blot pattern it appears that the SK4 channel is widely expressed in the heart. Your model of action of SK4 blockade mostly discusses its role in SAN pacemaker cells. Since however SK4 is expressed throughout the heart, we also need to take into account its role in working myocytes.
3. The legends of the figures are far too long and also repeating the results description made in the body of the text.

Referee #3 (Remarks):

The manuscript by Haron-Khun et al suggests the potassium channel SK4 as novel potential target for the treatment of CPVT associated tachyarrhythmias. The paper is based upon a previous study from the same group which demonstrated expression on SK4 in human embryonic stem cell derived cardiomyocytes. The authors now show expression of SK4 in human iPS derived cardiomyocytes from healthy controls and patients with CPVT2 as well as in primary mouse cardiomyocytes from WT and transgenic mice expressing the same calsequestrin mutation as the patients. In both cases the SK4 inhibitor TRAM-34 reduced DADs following stimulation with ISO, while in vivo EEG recordings demonstrate that TRAM-34 and clotrimazole at 20 mg/kg reduce arrhythmia in CASQ2-D307H knockouts and full CAQ2 knockout mice. Overall, these are very interesting findings that are potentially of high clinical relevance.

As a reviewer I have the following suggestions for improving the study.

1. Why do the authors need 5 microM TRAM-34 to block the KCa current in Fig 1 a/b and Fig 2a/b? TRAM-34 has a reported IC50 of 20 nM for SK4? Why don't the authors use less? 100 nM, 500 nM

or 1 microM? TRAM-34 starts blocking multiple KV channels as well as Nav channels at concentrations of 5 to 10 microM. The experiments would be a lot "cleaner" with lower concentrations. Do the authors apply TRAM-34 through a perfusion system with a lot of plastic tubing? TRAM-34 is notoriously sticky.

2. Why do the authors have to use such different ISO concentrations for the iPS derived cardiomyocytes in Fig 1 and the primary mouse myocytes in Fig 2? 3 microM ISO versus 50 nM ISO is a huge difference.

3. It would be fairer to show TRAM-34 sensitive current in Fig 1C and 2C as a scatter plot including the cells that did not express detectable current. The authors state in the text that 7 out of 15 normal iPS-CM cells showed SK4 current and 9 out of 13 CPVT2 derived cells. The bar graphs in Fig 1C and 2C only seem to show the data from the positive cells.

4. Please specify how long after telemetry lead implantation the mice were used for the EEG experiments. The methods state at least 24 hours, which is very short. Most laboratories will allow animals to recover 7 to 14 days to avoid any effects of inflammation from the surgical procedure. Considering that TRAM-34 is an anti-inflammatory drug that could play a role for the interpretation of the results.

1st Revision - authors' response

10 January 2017

Reviewer 1

We thank this reviewer for her/his constructive and insightful comments and advices. All the changes in the revised manuscript have been labeled in red to facilitate the reviewing. To address the concerns of the three reviewers and clarify a number of issues, we have added new experiments, including: series of new voltage-clamp experiments to isolate the SK4 currents in human induced pluripotent stem cells (hESC-CMs) and in sinoatrial cells (SAN) using a lower concentration of TRAM-34 (1 μ M) as well as new current-clamp experiments in hESC-CMs with 1 μ M TRAM-34. To further provide a mechanistic insight into how SK4 currents (ISK4) contribute to SAN pacemaker activity, the impact of ISK4 in SAN firing rate was examined using mathematical modelling (New Appendix Fig. S5). As requested, we also provided an appropriate statistical data processing for electrophysiological results of both hESC-CMs and SAN cells.

1- The study by Haron-Khun et al. describes SK4 block as a novel putative therapeutic strategy to treat Catecholaminergic Polymorphic Ventricular Tachycardia (CPVT). Therefore the authors studied iPSCs and Knock-In mouse models of CPVT and particularly focused on the role of SK4 sino-atrial node (SAN) cells and the effects of SK blockers in mouse ECG recordings. Concerning the mechanism of action for the rescue of the CPVT phenotype by SK4 channel block the authors state: Because of this slow channel deactivation, we suggest that SK4 channel contribution becomes significant only at the late repolarization, thereby contributing to the MDP hyperpolarization, which facilitates activation of If and recovery from inactivation of voltage gated Ca²⁺ channels. Thus, the net effect of SK4 channel activation will be an increase in the firing rate (manuscript in preparation). As this not provided data describes the molecular mechanism of action and the principle how SK4 channel block alters/rescues the electrical properties of cardiac cells, some of the data must be moved into the current EMBO Mol Med manuscript to undermine the counter intuitive idea that sino-atrial SK4 expression is in fact increasing excitability.

We thank this reviewer for providing us with the opportunity to add both modeling and experimental data as requested. To further explore the mechanistic insight into how SK4 currents (ISK4) contribute to SAN pacemaker activity, the impact of ISK4 in SAN firing rate was examined using mathematical modelling, where ISK4 was added to the mouse model implemented by Kharche et al. (Kharche et al., 2011). Appendix Figure S5 shows the model predictions with and without the contribution of ISK4. From the Ca²⁺-dependent sensitivity curve of SK4 channel activation measured by Logsdon et al. (Logsdon et al., 1997), we constrained the model with a Hill slope of $n_x = 2.7$ and a Ca²⁺ dissociation constant of $k_x = 0.27 \mu$ M. Assuming activation and deactivation time constants to $\tau_a = 5$ ms and $\tau_d = 50$ ms, respectively, as referred for all SK channels (Berkefeld et al., 2010), addition of ISK4 resulted in a slower AP upstroke in late DD (leading to a decrease in the firing rate) and in a faster AP repolarization (leading to an increase in the firing rate). Because of this time delay in channel activation and deactivation processes, the net effect of adding ISK4 to the model resulted in an increase in the firing rate (Appendix Fig. S5A and B). ISK4 is still active even after calcium concentration in the membrane subspace [Ca²⁺]_{sub} returned to its basal value (Appendix Fig. S5C and D). This is mainly due to the time constant of the deactivation process. By keeping the values of $n_x = 2.7$ and $k_x = 0.27 \mu$ M, but removing from the equation the activation and deactivation

time constants, the effect of adding ISK4 was opposed to the experimental findings as the firing rate decreased (Appendix Fig. S5E and F). Because no time delay occurs in the activation and deactivation processes, the contribution of ISK4 during late DD outweighs its effect during late repolarization. This leads ISK4 to vanish very quickly after the peak of the AP and to follow the $[Ca^{2+}]_{sub}$ trajectory (Appendix Fig. S5G and H).

Because of this slow channel deactivation, we suggest that SK4 channel contribution becomes significant only at the late repolarization, thereby contributing to the MDP hyperpolarization, which facilitates activation of I_f and recovery from inactivation of voltage-gated Ca^{2+} channels. Thus, the net effect of SK4 channel activation will be an increase in the firing rate. As shown in the new Appendix Fig. S2, blocking SK4 channels with clotrimazole or TRAM-34 will significantly decrease the firing rate and increase the MDP.

2- Fig. 1. The TRAM-sensitive current was present in 7 out of 15 (control) and 9 out of 13 cells (CPVT). As this data describes the SK4 current, the analyses is not done careful enough. The average TRAM-sensitive current density (Fig. 1c) should include all cells and not only the cells in which the experiments apparently worked and the authors isolated a relative large TRAM-sensitive current.

We thank the reviewer for this judicious comment. Accordingly, we present now the TRAM-34-sensitive current densities of hESC-CMs from normal and CPVT2 patients by a scatter plot incorporating data of hESC-CMs that were insensitive (zero currents) and sensitive to 5 μ M TRAM-34 (Fig. 1C). In addition, we made new experiments (5 cells) where we measured the TRAM-34-sensitive currents using 1 μ M TRAM-34 for purpose of selectivity concerns. We found appropriate to add these data to the scatter plots. Similar TRAM-34-sensitive current densities were found using either 1 μ M or 5 μ M TRAM-34 (Fig. 1C). No significant differences were found in TRAM-34-sensitive current densities of normal and CPVT2 hiPSC-CMs (Fig. 1C).

For selectivity purposes, we examined whether TRAM-34 interfered with major pacemaker currents in hESC-CMs. We found that 5 μ M TRAM-34 did not alter T type and L- type Ca^{2+} currents measured by the two inward humps (zero free Ca^{2+} in pipet solution; Appendix Fig. S1A). While 25 μ M ZD7288 blocked I_f at all voltages ($\sim 70\%$ inhibition at -100 mV), 5 μ M TRAM-34 did not affect the I_f current at any voltage. The NCX blocker KB-R7943 (3 μ M), potentially inhibited the NCX current, but 5 μ M TRAM-34 was ineffective (Appendix Fig. S1B and C).

3-In addition, the authors should illustrate the TRAM-sensitive current. Here the average of the TRAM-sensitive current would make sense to be illustrated in Fig. 1a and Fig. 1b. This additional data will show the rectification properties and reversal potential of the putative SK4 current.

We have illustrated the average of the TRAM-sensitive currents from the new experiments where 1 μ M TRAM-34 was used. The representative traces shown in new Fig. 1A and B correspond to those using 1 μ M TRAM-34. Subtracting the ramp currents in solution 1 to those in solution 1+TRAM-34 (1 μ M) yielded the TRAM-34-sensitive current. Figure 1D shows the average traces of the TRAM-34-sensitive currents (using 1 μ M TRAM-34) of normal and CPVT2-derived hiPSCs, which mainly exhibited an outward component. Yet, small residual inward currents likely corresponding to cationic conductances were not fully blocked by solution 1 and therefore shifted the E_{rev} to values more positive than those of EK.

4-Why is the control data not illustrated for the DD slope in Fig. 1e, while it is always provided (for Rate, APD50, Number of DADs..)? Fig. 1e: The authors state "Adding 5 μ M TRAM-34 depolarized the maximal diastolic potential (MDP)". As this is the major mechanism of action, as argued in the Discussion section (see above), the authors should provide the data and statistics in Fig. 1e.

These forgotten important data are now provided in the new Fig. 2B.

5-Fig. 2a-c. There are problems with the analyses of data display of the TRAM-sensitive current in SAN cells. The representative traces in Fig. 1a suggest a huge TRAM-sensitive current in comparison to the recordings of the SAN KI (Fig. 1b). This cannot be representative as the analyses in Fig. 1c suggest that there is no difference in current densities. As for Fig. 1, the average TRAM-sensitive currents should be displayed. Moreover no information is provided whether also these cells, as for the iPSCs, did not always have a TRAM-sensitive current. As stated above the statistics should be done using all cells and not only the "responders" to get an impression of the SK4 current amplitude in this cardiac tissue.

We thank the reviewer for this judicious comment. As for hESC-CMs, we have now illustrated the representative traces shown in new Fig. 3A and B with those using 1 μM TRAM-34.

For the average of the TRAM-sensitive currents we show those from the new experiments where 1 μM TRAM-34 was used (new Fig. 3D).

We also present now the TRAM-34-sensitive current densities of SAN cells from WT and CASQ2KI mice by a scatter plot incorporating data of SAN cells that were insensitive (zero currents) and sensitive to 5 μM TRAM-34 (Fig. 1C). In addition, we made new experiments (7 cells), where we measured the TRAM-34-sensitive currents using 1 μM TRAM-34 for purpose of selectivity concerns. We found appropriate to add these data to the scatter plots. Similar TRAM-34-sensitive current densities were found using either 1 μM or 5 μM TRAM-34 (Fig. 3C). No significant differences were found in TRAM-34-sensitive current densities of SAN cells from WT and CASQ2KI mice (Fig. 3C).

6-Also Fig. 2e lacks the analyses of the maximal diastolic potential.

This is now provided in new Fig. 4B.

7-The authors should consider in the Discussion/Results section that a PQ interval prolongation could be also caused by atrial effects and does not strictly indicate a role of SK4 in the conduction

system

We thank this reviewer for this judicious comment. Accordingly, we have added the following sentence in the discussion.” The prolongation of the PR interval is usually related to either AV node and/or the His–Purkinje system and suggests that SK4 channels are expressed in the conduction system. However, the PR interval represents a composite of several components. A prolonged PR interval can also reflect delayed interatrial conduction times. Prolonged PR interval was often considered detrimental to diastolic filling because it leads to a decrease in diastolic filling time. However, a prolonged PR interval could be also beneficial, because it may allow for complete atrial emptying during the atrial systole.”

Reviewer 2

We thank this reviewer for her/his constructive and insightful comments and advices. All the changes in the revised manuscript have been labeled in red to facilitate the reviewing. To address the concerns of the three reviewers and clarify a number of issues, we have added new experiments, including: series of new voltage-clamp experiments to isolate the SK4 currents in human induced pluripotent stem cells (hESC-CMs) and in sinoatrial cells (SAN) using a lower concentration of TRAM-34 (1 μ M) as well as new current-clamp experiments in hESC-CMs with 1 μ M TRAM-34. To further provide a mechanistic insight into how SK4 currents (ISK4) contribute to SAN pacemaker activity, the impact of ISK4 in SAN firing rate was examined using mathematical modelling (New Appendix Fig. S5). As requested, we also provided an appropriate statistical data processing for electrophysiological results of both hESC-CMs and SAN cells.

1-In the Western blot shown in Fig. 2d it appears that the expression levels of SK4 are affected by the CPVT phenotype and are reduced in the SAN and left and right ventricles. Is this impression correct and have you quantified your Western blots? If that is indeed the case, it would indicate that the CPVT disease process may have impact on SK4 channel expression.

We thank this reviewer for the judicious remark. We quantified the different Western blots. Quantitative analysis of the blots showed no significant differences in the heart tissues between the WT and CASQ2-D307H KI mice (see new Fig. 3E and F).

2-The slowing of the PR interval after TRAM-34 blockade suggests that not only is there expression in the SAN but also in the AV node. In the discussion you propose that expression is present throughout the cardiac conduction tissue. This could be demonstrated by performing immunohistochemical stainings provided that your antibody not only works in Western blots but also on tissue sections. Based on the Western blot pattern it appears that the SK4 channel is widely expressed in the heart. Your model of action of SK4 blockade mostly discusses its role in SAN pacemaker cells. Since however SK4 is expressed through the heart, we also need to take into account its role in working myocytes.

This is a very interesting point. We tried very hard to perform immunohistochemical staining on mouse heart tissue sections, using two different anti-SK4 antibodies, but we failed. These antibodies were very good for Western blots but were inefficient for immunostaining at least in our hands. As for the presence of SK4 channels in atria and in relation to the prolonged PR interval, we have added in the discussion the following sentence:”The prolongation of the PR interval is usually related to either AV node and/or the His–Purkinje system and suggests that SK4 channels are expressed in the conduction system. However, the PR interval represents a composite of several components. A prolonged PR interval can also reflect delayed interatrial conduction times. Prolonged PR interval was often considered detrimental to diastolic filling because it leads to a decrease in diastolic filling time. However, a prolonged PR interval could be also beneficial, because it may allow for complete atrial emptying during the atrial systole.”

3-The legends of the figures are far too long and also repeating the results description made in the body of the text.

We agree with the reviewer and have significantly shorten all figure legends of the manuscript.

Reviewer 3

We thank this reviewer for her/his constructive and insightful comments and advices. All the changes in the revised manuscript have been labeled in red to facilitate the reviewing. To address the concerns of the three reviewers and clarify a number of issues, we have added new experiments, including: series of new voltage-clamp experiments to isolate the SK4 currents in human induced pluripotent stem cells (hESC-CMs) and in sinoatrial cells (SAN) using a lower concentration of TRAM-34 (1 μ M) as well as new current-clamp experiments in hESC-CMs with 1 μ M TRAM-34. To further provide a mechanistic insight into how SK4 currents (ISK4) contribute to SAN pacemaker activity, the impact of ISK4 in SAN firing rate was examined using mathematical modelling (New

Appendix Fig. S5). As requested, we also provided an appropriate statistical data processing for electrophysiological results of both hESC-CMs and SAN cells.

1. Why do the authors need 5 microM TRAM-34 to block the KCa current in Fig 1 a/b and Fig 2a/b? TRAM-34 has a reported IC50 of 20 nM for SK4? Why don't the authors use less? 100 nM, 500 nM or 1 microM? TRAM-34 starts blocking multiple KV channels as well as Nav channels at concentrations of 5 to 10 microM. The experiments would be a lot "cleaner" with lower concentrations. Do the authors apply TRAM-34 through a perfusion system with a lot of plastic tubing? TRAM-34 is notoriously sticky.

We thank the reviewer for this important comment and agree with him. Accordingly, we performed a series of new experiments including: series of new voltage-clamp experiments to isolate the SK4 currents in human induced pluripotent stem cells (hESC-CMs) and in sinoatrial cells (SAN) using a lower concentration of TRAM-34 (1 μ M) as well as new current-clamp experiments in hESC-CMs with 1 μ M TRAM-34 (new figures 1, 2 and 3). The results were very similar to those using 5 μ M TRAM-34 as show in the scatter plots in new Figs 1C and 3C. We also provided the representative traces as well as the average of TRAM-34-sensitive currents using 1 μ M TRAM-34 (new Fig. 1A,B and D; new Fig. 3A,B and D). See the example of new Fig. 1.

For selectivity purposes, we examined whether TRAM-34 interfered with major pacemaker currents in hESC-CMs. We found that 5 μ M TRAM-34 did not alter T type and L- type Ca²⁺ currents measured by the two inward humps (zero free Ca²⁺ in pipet solution; Appendix Fig. S1A). While 25 μ M ZD7288 blocked If at all voltages (~70 % inhibition at -100 mV), 5 μ M TRAM-34 did not affect the If current at any voltage. The NCX blocker KB-R7943 (3 μ M), potently inhibited the NCX current, but 5 μ M TRAM-34 was ineffective (Appendix Fig. S1B and C).

2. Why do the authors have to use such different ISO concentrations for the iPS derived cardiomyocytes in Fig 1 and the primary mouse myocytes in Fig 2? 3 microM ISO versus 50 nM ISO is a huge difference.

We totally agree with the reviewer. Accordingly, together with lower concentrations of TRAM-34, we used lower concentration of isoproterenol at 100 nM (see new Fig. 2).

3-It would be fairer to show TRAM-34 sensitive current in Fig 1C and 2C as a scatter plot including the cells that did not express detectable current. The authors state in the text that 7 out of 15 normal iPS-CM cells showed SK4 current and 9 out of 13 CPVT2 derived cells. The bar graphs in Fig 1C and 2C only seem to show the data from the positive cells.

We thank the reviewer for this judicious remark. We have now provided scatter plots in new Fig. 1C and new Fig. 3C, which show TRAM-34 sensitive currents, including cells that were insensitive to TRAM-34 (see the Fig. 1 above).

4-Please specify how long after telemetry lead implantation the mice were used for the EEG experiments. The methods state at least 24 hours, which is very short. Most laboratories will allow animals to recover 7 to 14 days to avoid any effects of inflammation from the surgical procedure. Considering that TRAM-34 is an anti-inflammatory drug that could play a role for the interpretation of the results.

We agree the reviewer that the sentence was not clear and not precise enough. In fact, animals were allowed to recover after surgery 5-6 days before any experiments (see methods). This time allowed perfect recovery from a careful surgery.

2nd Editorial Decision

20 January 2017

Thank you for the submission of your revised manuscript to EMBO Molecular Medicine. We have now received the enclosed reports from the referees that were asked to re-assess it. As you will see the reviewers are now globally supportive and I am pleased to inform you that we will be able to accept your manuscript pending the following final amendments:

- 1) Due to production restrictions, the table must be provided in black and white
- 2) Please remove the colored lettering from the manuscript and Appendix file as it is no longer needed.

- 3) Please confirm that the figures in the point-by-point rebuttal can be included in the peer review process document to be published along side the manuscript.
- 4) As per our Author Guidelines, the description of all reported data that includes statistical testing must state the name of the statistical test used to generate error bars and P values, the number (n) of independent experiments underlying each data point (not replicate measures of one sample), and the actual P value for each test (not merely 'significant' or ' $P < 0.05$ ').
- 5) We encourage the publication of source data, with the aim of making primary data more accessible and transparent to the reader. Would you be willing to provide a PDF file per figure that contains the original, uncropped and unprocessed scans of all or at least the key gels used in the manuscript and/or source data sets for relevant graphs? The files should be labeled with the appropriate figure/panel number, and in the case of gels, should have molecular weight markers; further annotation may be useful but is not essential. The files will be published online with the article as supplementary "Source Data" files. If you have any questions regarding this just contact me.
- 6) Every published paper includes a 'Synopsis' to further enhance discoverability. Synopses are displayed on the journal webpage and are freely accessible to all readers. They include a short standfirst as well as 2-5 one sentence bullet points that summarise the paper. Please provide the synopsis including the short list of bullet points that summarise the key NEW findings. The bullet points should be designed to be complementary to the abstract - i.e. not repeat the same text. We encourage inclusion of key acronyms and quantitative information. Please use the passive voice. Please attach this information in a separate file or send them by email, we will incorporate it accordingly. You are also welcome to suggest a striking image or visual abstract to illustrate your article. If you do please provide a jpeg file 550 px-wide x 400-px high.

Please submit your revised manuscript within two weeks. I look forward to seeing a revised form of your manuscript as soon as possible.

***** Reviewer's comments *****

Referee #1 (Remarks):

The manuscript is strongly improved and I have no additional/further comments

Referee #2 (Remarks):

The authors were responsive to the reviewer's comments and have revised the manuscript accordingly.

Referee #3 (Remarks):

I have no further comments. The authors have addressed all my previous concerns satisfactorily. I now believe that SK4 plays a role in the heart.

Corresponding Author Name: Bernard Attali

Journal Submitted to: Embo Molecular Medicine

Manuscript Number: EMM-2016-06937